# Dual Hyaluronic Acid and Folic Acid Targeting pH-Sensitive Multifunctional 2DG@DCA@MgO-Nano-Core–Shell-Radiosensitizer for Breast Cancer Therapy

**DOI:** 10.3390/cancers13215571

**Published:** 2021-11-07

**Authors:** Mostafa A. Askar, Noura M. Thabet, Gharieb S. El-Sayyad, Ahmed I. El-Batal, Mohamed Abd Elkodous, Omama E. El Shawi, Hamed Helal, Mohamed K. Abdel-Rafei

**Affiliations:** 1Radiation Biology Department, National Center for Radiation Research and Technology (NCRRT), Egyptian Atomic Energy Authority, Cairo 11787, Egypt; noura.m.thabet@eaea.org.eg (N.M.T.); mohamed.abdelrafei@eaea.org.eg (M.K.A.-R.); 2Drug Microbiology Laboratory, Drug Radiation Research Department, National Center for Radiation Research and Technology (NCRRT), Egyptian Atomic Energy Authority, Cairo 11787, Egypt; Ahmed.Elbatal@eaea.org.eg; 3Department of Electrical and Electronic Information Engineering, Toyohashi University of Technology, Toyohashi 441-8580, Japan; mohamed.hamada.abdlekodous.xi@tut.jp; 4Health and Radiation Research Department, National Center for Radiation Research and Technology (NCRRT), Egyptian Atomic Energy Authority, Cairo 11787, Egypt; dromaymaelshawi@hotmail.com; 5Zoology Department, Faculty of Science, Al-Azhar University, Cairo 11651, Egypt; helal@azhar.edu.eg

**Keywords:** breast cancer, targeted therapy, multifunctional core–shell nanoparticles, antitumor, radio-sensitization

## Abstract

**Simple Summary:**

In this study, we have developed CD44 and folate receptor-targeting multi-functional dual drug-loaded nanoparticles. This comprises hyaluronic acid (HA) and folic acid (FA) conjugated to 2-deoxy glucose (2DG) and a shell linked to a dichloroacetate (DCA) and magnesium oxide (MgO) core (2DG@DCA@MgO; DDM) to enhance the localized chemo-radiotherapy for effective breast cancer (BC) treatment. The physicochemical properties of nanoparticles including stability, selectivity, responsive release to pH, cellular uptake, and anticancer efficacy were comprehensively examined. Mechanistically, we identified multiple component signal pathways as important regulators of BC metabolism and mediators for the inhibitory effects exerted by DDM. Nanoparticles exhibited sustained DDM release properties in bio-relevant media, which was responsive to acidic pH providing edibility to the control of drug release from nanoparticles. DDM-loaded and HA–FA-functionalized nanoparticles exhibited increased selectivity and uptake by BC cells. Cell-based assays indicated that the functionalized DDM significantly suppressed cancer cell growth and boosted radiotherapy (RT) efficacy via inducing cell cycle arrest, enhancing apoptosis, and modulating glycolytic and OXPHOS pathways. Accordingly, the inhibition of glycolysis/OXPHOS by DDM and RT treatment may result in cancer metabolic reprogramming via a novel PI3K/AKT/mTOR/P53NF-κB/VEGF pathway in BC cells. Therefore, the dual targeting of glycolysis/OXPHOS pathways is suggested as a promising antitumor strategy.

**Abstract:**

Globally, breast cancer (BC) poses a serious public health risk. The disease exhibits a complex heterogeneous etiology and is associated with a glycolytic and oxidative phosphorylation (OXPHOS) metabolic reprogramming phenotype, which fuels proliferation and progression. Due to the late manifestation of symptoms, rigorous treatment regimens are required following diagnosis. Existing treatments are limited by a lack of specificity, systemic toxicity, temporary remission, and radio-resistance in BC. In this study, we have developed CD44 and folate receptor-targeting multi-functional dual drug-loaded nanoparticles. This composed of hyaluronic acid (HA) and folic acid (FA) conjugated to a 2-deoxy glucose (2DG) shell linked to a layer of dichloroacetate (DCA) and a magnesium oxide (MgO) core (2DG@DCA@MgO; DDM) to enhance the localized chemo-radiotherapy for effective BC treatment. The physicochemical properties of nanoparticles including stability, selectivity, responsive release to pH, cellular uptake, and anticancer efficacy were thoroughly examined. Mechanistically, we identified multiple component signaling pathways as important regulators of BC metabolism and mediators for the inhibitory effects elicited by DDM. Nanoparticles exhibited sustained DDM release properties in a bio-relevant media, which was responsive to the acidic pH enabling eligibility to the control of drug release from nanoparticles. DDM-loaded and HA–FA-functionalized nanoparticles exhibited increased selectivity and uptake by BC cells. Cell-based assays revealed that the functionalized DDM significantly suppressed cancer cell growth and improved radiotherapy (RT) through inducing cell cycle arrest, enhancing apoptosis, and modulating glycolytic and OXPHOS pathways. By highlighting DDM mechanisms as an antitumor and radio-sensitizing reagent, our data suggest that glycolytic and OXPHOS pathway modulation occurs via the PI3K/AKT/mTOR/NF-κB/VEGF_low_ and P53_high_ signaling pathway. In conclusion, the multi-functionalized DDM opposed tumor-associated metabolic reprogramming via multiple signaling pathways in BC cells as a promising targeted metabolic approach.

## 1. Introduction

Breast cancer (BC) is one of the most common malignant tumors with the highest mortality-rate in women. Therapeutic options for BC chemotherapy have several limitations, such as a lack of therapeutic efficacy, toxicity toward healthy tissues, and high rate of metastasis and recurrence [1,2]. Accordingly, new therapeutic strategies must be developed to target and inhibit BC growth, which is characterized by higher glycolytic activity, glutamine consumption, glutaminolysis levels generated via aerobic conditions (Warburg effect) and mitochondria oxidative phosphorylation (OXPHOS), and the overexpression of CD44 receptors and folate receptors-α (FR-α) [3,4,5]. Recent studies indicated that the tumor suppressor, P53, phosphatidylinositol 3-kinase (PI3K), protein kinase B (AKT), mammalian target of rapamycin (mTOR), nuclear factor-κ light chain enhancer of activated B cells (NF-κB), and vascular endothelial factor (VEGF) pathways are mutated and related to a wide variety of cancer cell phenotypes, including uncontrolled proliferation, genomic instability, and metabolic reprogramming [6]. Therefore, aerobic glycolysis, mitochondrial metabolism, CD44, and FR-α have been considered as being particularly important for the selection and targeting by anticancer drugs. The exploration of drug therapy combinations is a promising strategy, as synergistic effects from multiple drugs potentially lead to better therapeutic outcomes and a reduced probability of drug resistance in cancer cells [7]. 2-Deoxy glucose (2-DG) and dichloroacetate (DCA) are drug combinations currently used in cancer clinical trials [8]. Both 2-DG and DCA are established as BC chemotherapeutic treatments and have been combined with radiotherapy (RT) in clinical trials. However, the clinical use of DCA is restricted due to mild toxicity in hematologic, hepatic, renal, and cardiac systems [9,10]. Recently, MgO-NPs have been highlighted for their multi-potential activity as drug delivery, anticancer, radio-sensitization, magnetic resonance imaging, and hyperthermia systems, and the incorporation of MgO-NPs in nano materials could help track drug uptake by TEM imaging and IR quantitative measurements. Moreover, MgO in nano materials core can be formed regularly crystal in fact and produced the face-centered cubic (FCC) crystalline configuration in nanoparticles fabrication [11,12]. Similarity, these treatments are also still limited due to rapid relapse issues, systemic toxicity, and non-specific drug delivery, leading to low therapeutic outcomes [13]. Additionally, poor survival rates in patients with BC may be attributable to radio-resistance stemming from increased cell DNA repair capabilities after RT [14]. Since tumor tissues constitute an acidic environment and the intracellular compartments of cancer cells (endosomes and lysosomes) provide an even lower pH than the extracellular environment (around 5.0–5.5), the pH-responsive release behavior of nanoparticles is potentially important with regard to the in vivo anticancer efficacy. Although it is generally desirable to lower glucose levels in malignant cells, it should be recognized that administration of glucose may lower tumor cell pH and thus facilitate drug delivery [5]. Therefore, the development of strategies to overcome these limitations and provide targeted and controlled release for efficient chemo-radiotherapy in BC patients is essential [15]. Nanomedicines have served as vital drug delivery systems and improved cancer therapy with reduced systemic toxicity of anticancer drugs in healthy tissues [16]. The drug encapsulated in nanocarriers are promising modalities as they increase drug accumulation at tumor tissues due to enhanced permeability and retention (EPR) effects, thereby increasing cancer therapeutics efficacy [17]. Because of these advantages, we have developed a stimuli-reacting core–shell nanoparticle system conjugated with hyaluronic acid (HA) and folic acid (FA), consisting of two different agents for multi-targeted BC therapy. Moreover, the DDM was surface conjugated with HA and FA to target CD44 and FR-α receptors, which are overexpressed in BC cells. Similarity, CD44 and folate receptors have a high affinity for HA and FA, which is usually captured to nourish fast-dividing BC cells [18]. Therefore, our design innovation was underpinned by a novel biodegradable multi-target nanocarrier that targets the overexpressed CD44 and FR-α receptors in BC cells. This is followed by the release of the three therapeutic agents for the effective chemo-radiotherapy of BC.

## 2. Materials and Methods

### 2.1. Synthesis of DDM

#### 2.1.1. Preparation of MgO Nanoparticles

Firstly, MgO nanoparticles (MgO-NPs) were prepared by the method reported by Diana et al. [19]. Briefly, urea was added to an aqueous solution of magnesium nitrate (0.25 M) under constant stirring at 70 °C until gel formation. Then, MgO-NPs were formed by placing the gel at 500 °C using a muffle furnace for 3 h. Finally, formed MgO-NPs were washed with deionized water (D.I.W) and dried at 70 °C for 2 h (Figure 1a).

#### 2.1.2. Preparation of MgO@DCA@2DG

This novel composite structure was prepared by a simple impregnation method. Firstly, prepared MgO-NPs (from step 1) were dispersed in 50 mL ethanol using water bath sonication for 45 min. Then, aqueous solutions of previously prepared DCA and 2DG (10 mM) were added to be dispersed under constant stirring for 2 h at room temperature (25 °C). Finally, the formed powder was collected, washed, and dried (Figure 1a).

#### 2.1.3. Preparation of MgO@DCA@2DG Conjugated with HA and FA

The formed composite structure (from step 2), was dispersed in D.I.W by water bath sonication for 30 min. Then, aqueous solutions of HA and FA (20 mM) were added into the solution, followed by sonication of the mixture for 30 min. Third, the mixture was left to stirrer for 2 h at constant stirring. Finally, the resulted powder was collected using centrifugation and dried at 80 °C for 1 h (Figure 1a).

### 2.2. Characterization of DDM

Firstly, the stoichiometry of the synthesized DDM is examined via by employing the energy-dispersive X-ray spectra ((EDX), JEOL JSM-5600 LV, Tokyo, Japan). To confirm the formation of the exact sample with detected functional groups, Fourier transform infrared (FT-IR) spectroscopy (NICOLET iS10 model instrument, Tokyo, Japan) was conducted over a wide range (400–4000 cm^−1^). The crystal structure of the samples was investigated via the x-ray diffraction technique (XRD; Shimadzu XRD-6000). XRD patterns were obtained in the range of 2θ from 17° to 90° at room temperature. Cu Kα was used as a radiation source of wavelength λ = 0.15408 nm, scan rate 0.8°/min, operation voltage 50 kV, and current 40 mA [20,21]. Information on the surface morphology of the samples’ particles was obtained using scanning electron microscopy ((SEM), JEOL JSM-5600 LV, Tokyo, Japan). Finally, the shape and size of the synthesized samples were obtained by a high resolution Transmission electron microscopy ((HRTEM), JEOL JSM-5600 LV, Tokyo, Japan).

### 2.3. Stability of DDM

To determine stability, DDM charge and sizes were studied on incubation with PBS and 10% fetal bovine serum (FBS; pH 7.4) at body temperature (37 °C) using DLS for 6 days [5].

### 2.4. Cell Culture

The cell lines used in this study were purchased from the Cell Culture Department, VACSERA (Cairo, Egypt). Normal cells (MCF-10A), MCF-7, and MDA-MB-231 BC cells were cultured in Dulbecco’s modified Eagle’s medium (DMEM) supplemented with penicillin–streptomycin (100 U/mL) and 10% fetal bovine serum (FBS) in a 5% CO_2_ humidified chamber at 37 °C. For different study treatments, cells were used at 100% confluency.

### 2.5. Cytotoxicity/Morphology Assay

The cytotoxic effects of dual drug-loaded nanoparticles (DDM) were analyzed using the 3-(4,5-Dimethylthiazol-2yl)-2,5-diphenyltetrozolium bromide (MTT) assay (Sigma-Aldrich, St. Louis, MO, USA) [22]. DDM was dissolved in dimethyl sulfoxide (DMSO) (Sigma). The stock solutions were diluted with a culture medium to the indicated concentration for treatment before usage and the final concentration of DMSO in each well was 0.01% (*v*/*v*). Cells treated with the vehicle only were kept as the control group. Briefly, BC cells (1 × 10^5^ cells/mL (100 µL/well) were seeded in 96-well plates and cultured for 24 h. The medium was then replaced by 5 mg MTT in 20 µL DMEM. Cells were further incubated for 4 h, the DMEM/MTT mixture removed, and 150 μL DMSO added to dissolve formazan crystals. Then, the absorbance was measured using an enzyme-linked immunosorbent assay (ELISA) plate reader (BioTeck, Bad Friedrichshall, Germany) at 570 nm. The half-maximal inhibitory concentration (IC_50_) was calculated using SPSS one-way ANOVA (IBM Inc., Chicago, IL, USA). Graphs were drawn using Graph-Pad Prism software version 8.0 (Graph-Pad Prism Inc., San Diego, CA, USA). Cell morphology was recorded using a phase-contrast inverted microscope fitted with a digital camera (Nikon, Japan). All studies were performed in triplicate.

### 2.6. DDM Release

To study the drug release, the DDM suspensions were subjected to pH 3, 7, and 9 at 37 °C conditions. At pre-determined time points, the particles were collected using an external magnet (1.3 Tesla), and the supernatant was saved for analysis after 24 h incubation. MgO release was quantified at 285.2 nm absorbance using UV-vis.

### 2.7. Cell Selectivity and DDM Uptake

Normal (MCF-10A), MCF-7, and MDA-MB-231 cells were seeded in 24-well plates with round coverslips at 2 × 10^4^ cells/well. In the next day, cells were incubated with a medium containing an IC_50_ DDM dose. After a 24 h incubation, the cells were washed three times in phosphate buffer saline (PBS) and divided into four aliquots for different methods; the first and second methods investigated DDM cell selectivity via measurement of CD44 expression by flow cytometry, as described in the flowcytometric analysis part, and FR-α expression using qRT-PCR, as described in the real time PCR part. The third method investigated MgO levels in normal and cancer cells using atomic absorption spectrophotometry (AAS) AAS model (AI-1200), which was used with an air-acetylene burner (slot length = 11 cm). Instrument settings: Wavelength = 285.2 nm, lamp current = 5 mA, slit width = 0.2 nm, air-flow = 1.8 L/min, and ignition-flow = 2.4 L/min. A standard MgO solution was prepared by serially diluting 1000 mg/L MgO stock solution (Scharlau Chemie) in D.I.W. After 24 h incubation, cells were washed three times in PBS, and then centrifuged at 3000 rpm for 5 min, and supernatants were aspirated into bottles. Supernatants and pellets were then diluted in D.I.W and homogenized before MgO determination [23]. The fourth method was investigated DDM cell uptake using TEM Imaging in normal and cancer cells. In brief, cells were fixed in 2.5% glutaraldehyde in 0.1 M PBS, pH 7.4 (Electron Microscopy Sciences, Hatfield, PA, USA) in plates for 1 h at room temperature. Afterwards, cells were scraped from plates, centrifuged at a low speed, and suspended in 2.5% glutaraldehyde. Samples were processed at the Egyptian atomic energy Authority by post-fixation in 1% osmium-tetroxide, rinsing in distilled water, and dehydration through a graded acetone series. Samples were embedded in epoxy resin, cut into 70-nm sections, and then analyzed and photographed using a JEOL 100CXII TEM [24].

### 2.8. γ-ray Irradiation

Cells were irradiated with γ-rays using a ^137^Cs source (Gamma-cell-40 Exactor; NCRRT, AEA, Cairo, Egypt). The dose rate was 0.012 Gy/s. Dosimetry was used for all the studies to ensure dose uniformity and dose rates were delivered using a Fricke reference standard dosimeter [25].

### 2.9. Multi-MTT Assay

MCF-7 and MDA-MB-231 cells were plated at a density of 1000 cells/well in 6-well tissue culture plates and treated with DDM at IC_50_ doses of 281.9 and 192.8 µg/mL, respectively for 24 h. Cells were then exposed to 3 Gy (single dose) or 6 Gy (fractionated dose; 3 Gy every 3 days) of ^137^Cs-radiation (Figure 2a). During the study period, the drug-free medium was replaced every 3 days. Finally, cells were incubated with MTT, and cell survival cells were measured as described. Cell morphology was also recorded as described. Survival curves were calculated, to derive the Dose-Modifying Factor (DMF); this was calculated at the iso-effect of survival fraction (SF) = 50% with radiation treatment alone respective to combined treatments [26]. To calculate the proliferation-survival, only the early exponential phase of cell growth was used in the following equation; survival = 2 − (*t* delay/*t* doubling time), t delay = the time to reach a specific absorption value of irradiated cells vs. control, and t doubling time = the time required for cells to double [27].

### 2.10. Protocol of the Study

To evaluate anti-proliferative and radio-sensitizing efficacy of DDM, six study groups were designed (1) Cells without treatment (vehicle treated; Control), (2) Cells treated with DDM only (DDM), (3) Cells exposed to a fractionated dose (6 Gy) of irradiation alone (FDR), (4) Cells treated with DDM 24 h before FDR (DDM + 6Gy-FDR). Cell survival was assessed at 24 h, 48 h, and 72 h compared with the control group, to verify some important hypotheses on the mechanisms of action of DDM with or without FDR.

### 2.11. Cell Cycle, Apoptosis, and CD44 Analysis Using Flow Cytometry

MCF-7 and MDA-MB-231 cells (3 × 10^5^ cells/well) were either untreated (control group) or treated with an IC_50_ DDM dose (DDM) for 24 h and exposed to 6Gy-FDR (RT). After 24 h incubation, cells were harvested, washed twice in ice-cold PBS, and fixed overnight in 70% ethanol at 4 °C. Then, cells were washed in PBS, collected by centrifugation, and stained with propidium iodide (PI) (50 µg/mL) for cell cycle analysis. An annexin-V fluorescein isothiocyanate (FITC) kit (Beckman Coulter, Marseille, France) was used to measure apoptosis. A FITC-conjugated anti-CD44 antibody (1:400, Cat. No: YKIX337.8, eBioscience) was incubated with cells for 30 min at 4 °C to assess CD44 levels. All staining was performed using a FACSCanto II flow cytometer followed by analysis using BD Accuri-C6 Plus software (Biosciences, CA, USA) [28].

### 2.12. Determination of Glucose, Lactate, and Hyaluronic Acid (HA) Metabolism

After the incubation of MCF-7 and MDA-MB-231 with DDM for 24 h and then exposing them to RT, the measurement of glucose, lactate, and HA metabolism in treated and vehicle-treated MCF-7 and MDA-MB-231 cells was performed. Cell supernatants were harvested and indices were measured using commercial kits (Cat. No: GAGO20 and MAK064 (Sigma-Aldrich) for glucose and lactate, respectively) and HA (Cat. No: 029-001, Corgenix, Inc., Peterborough Cambridgeshire, UK) using a spectrophotometer (V-630 Bio UV-Vis, JASCO, Easton, MD, USA). Glucose, lactate, and HA concentrations were determined at 540 nm, 570 nm, and 450 nm, respectively.

### 2.13. Analysis of Hexokinase (HK) and Pyruvate Dehydrogenase (PDH) Activities

Intracellular HK and PDH activity was evaluated with a spectrophotometer (V-630 Bio UV-Vis: JASCO, USA) using Quantification Kit, Cat. No: MAK091-1KT and MAK183-1KT, respectively according to the manufacturer’s instructions (Merck KGaA, Sigma-Aldrich, Darmstadt, Germany). The HK and PDH concentrations were determined with OD values at 450 nm.

### 2.14. RNA Isolation and Real Time PCR Analysis

RNA extraction and qRT-PCR Total RNA from MCF-7 and MDA-MB-231 cells in each group was extracted by Trizol Reagent (Thermo Fisher Scientific, Waltham, MA, USA). cDNA was obtained from total RNA using the PrimeScript™ RT (Table 1) reagent kit (Takara Bio, Inc., Otsu, Japan). The expression of mRNA was assessed by qRT-PCR, which was carried out in triplicate by an SYBR Premix Ex Taq™ kit (Takara Bio, Inc. Shiga, Japan) and an ABI 7900HT Real-Time PCR system (Thermo Fisher Scientific). The primers used are presented in Table 1. GAPDH was used to normalize the results of qRT-PCR and the comparative cycle threshold values (2–ΔΔCt) were adopted to analyze the final results.

### 2.15. Measurement of Intracellular PKM2, HIF-1α, PDK1, NF-κB, VEGF, and ROS Levels by ELISA Assay

Pyruvate kinase isozymes M2 (PKM2), Pyruvate Dehydrogenase Kinase 1 (PDK1), Hypoxia-inducible factor 1-alpha (HIF-1α), NF-κB, VEGF, and reactive oxygen species (ROS) levels were determined by using the markers assay kit (My Biosource, San Diego, CA, USA; Cat. No: MBS2505089, MBS078206, MBS282197, MBS2089167, MBS355343, and MBS166870 respectively) following a modification of the manufacturer’s protocol. Cells were seeded in 150-mm plates, treated as described above, collected, and homogenized (2 × 10^6^) in 100 μL of ice-cold water. Reactions were carried out following the manufacture´s protocol and absorbance was measured at 570 nm using an automatic micro-plate reader (Quant, BioTek Instruments, Inc., Winooski, VT, USA).

### 2.16. Western Blotting Analysis

MCF-7 and MDA-MB-231 cells were seeded at 4 × 10^5^/well in 6-well plates. After treatments, cells were lysed in lysis buffer plus 10 μL PMSF (100 mM added to 1 mL buffer, Solarbio, Beijing, China) on ice for 30 min. Cells lysates were separated using 10% sodium dodecyl sulfate-polyacrylamide electrophoresis gels, blotted using polyvinylidenedifluoride membranes which, and blocked in 5% skim milk in PBS plus 0.1% Tween 20 (TBST). Membranes were then incubated overnight with primary antibodies at 4 °C for 12 h. The following antibodies were used: PI3K, AKT, mTOR total and phosphorylated, P53, SIRT1, and SIRT3 rabbit polyclonal antibodies (1:1000). A β-actin rabbit polyclonal antibody (1:4000, Proteintech, Rosemont, IL, USA) was used as a loading control for normalization. A secondary anti-rabbit antibody conjugated to horseradish peroxidase (1:4000; Proteintech) was incubated with membranes for 1 h at room temperature. Protein bands were visualized using enhanced chemiluminescent reagent (Thermo Fisher Scientific). Band images were obtained and quantified using the Protein Simple Digital imaging system (Flour Chem R, San Jose, CA, USA).

### 2.17. Statistical Analysis

All experiments were carried out at least in triplicate and the results were expressed as the mean ± standard error (SEM). The statistical software package (SPSS Inc., Chicago, IL, USA) was used for analysis. Statistical significance between all groups was analyzed by using the *p* < 0.001, *p* < 0.01 and *p* < 0.05. Statistical analyses graphs were drawn using Prism, version 8 (Graph Pad Software, La Jolla, CA, USA).

## 3. Results

### 3.1. Characterization of DDM

XRD displays a true detection of the crystallinity and the combination of the exposed DDM sample (Figure 1a), because it explains the status of the atoms, size, and axes. XRD results of the DDM powder are presented in Figure 1b, and many peaks were recognized for MgO-NPs. Diffraction characteristics are displayed inside 2θ (degree) as 24.20, 32.60, 37.50, 58.93, 62.03, and 72.18 where some peaks represent the Bragg’s appearances (001), (111), (20), (220), (311), and (222) extensions in that position sequentially, which can be recorded to the levels of cubic MgO (JCPDS 75-0447).

This suggests that the MgO-NPs (core structure) were regularly crystal in fact and produced the FCC crystalline configuration. It must be stated that the amorphous peak at 17.35 (*) was due to outer organic shells (DCA, 2DG, HA, and FA). The composition of the synthesized DDM sample was analyzed by EDX (Figure 1c), where the presence of O, C, Cl, and Mg is confirmed and the existence of Mg and O atoms is confirmed for the validation of MgO-NPs core. Moreover, the presence of O, C, and Cl is attributed to the DA, 2DG, HA, and FA multi-shells structures in the synthesized sample.

To further illustrate the core–shell structural features of the DDM, elemental mappings have been carried out selectively to the synthesized DDM and the images are depicted in (Figure 1d). It is evident from these images that the elements Mg, C, Cl, and O exist that agree with the preceding EDX results. Furthermore, these elements are homogeneously distributed. From the obtained images, we can conclude that both Mg (blue color) and O (pink color) atoms are located in the same place within DDM structure which confirms the core structure.

The SEM image of the synthesized DDM is shown in (Figure 2a). The surface behavior reveals dark layers represent the outer shells (HA and FA) with remarkable smooth agglomerates that can be observed due to the occupation of a large number of layers at the grain boundary which could control the grain growth. Additionally, the bright particles represented the MgO-NPs core which confirms the promising core–shell structure.

The HR-TEM image of the core–shell structure of the synthesized DDM is displayed in (Figure 2b). The synthesized composite possesses a semi-spherical structure with diameter sizes ranging from 165.21–88.94 nm with an average size of 99.87 nm. It must be noted that the condensed particles were attributed to the core MgO-NPs (yellow circles); while the faint layers were corresponding to the shell layers of DDM which are entirely validated by color in mapping/SEM images.

Figure 2c reveals the FT-IR spectra of the synthesized DDM. For the present nanocomposite, the characteristic vibration peak observed at 680 cm^−1^ was assigned to the stretching mode of MgO (in the core) and other assigned peaks were for the shells formed and were in good agreement with the literature. After conducting a comparative FTIR analysis of bare MgO-NPs, a peak located at 3230 cm^−1^ has corresponded to the -OH stretching region, also another peak located at 617 cm^−1^ was assigned to the stretching mode of the Mg-O core, which slightly shifted as compared with Mg-O (680 cm^−1^) in the synthesized nanocomposite due to the absence of organic shells. After performing the FTIR of the bare MgO-NPs, and confirming the presence of functional groups represented in the synthesized nanocomposite, this indicated the successful formation of the core–shell construction.

### 3.2. DDM Stability

To better understand nanoparticle stability, nanoparticle sizes were monitored by dynamic light scattering (DLS) (Figure 2d). The average hydrodynamic diameter of DDM remained essentially stable; nanoparticles did not aggregate over 6 days in PBS plus 10% FBS. DLS data indicated that DDM had a hydrodynamic diameter range of 237.8 ± 17 nm to 484 ± 26 nm. Moreover, with increased incubation time, the ζ-potential of nanoparticles was stabilized at values around −4.3 mV and −5.06 mV. Owing to the interactions of nanoparticle with media cationic constituents, it led to the neutralization of surface negative charges on nanoparticles, resulting in less negative ζ-potential values. Importantly, interactions did not generate nanoparticles aggregation even after a prolonged 6-day incubation.

### 3.3. The Anti-Proliferative Effects of DDM on BC Cell Growth

Anticancer activity results from MCF-7 and MDA-MB-231 cells indicated that DDM demonstrated anti-proliferative activities after 24 h. DDM proved a profound efficiency at most concentrations. The obtained IC_50_ was 281.9 µg/mL for MCF-7 cells and 192.8 μg/mL for MDA-MB-231 cells. The inhibitory effect of DDM was stronger in MDA-MB-231 cells than MCF-7 by less than half. Thus, DDM exhibited higher anticancer activity, especially toward triple-negative MDA-MB-231 cells (Figure 3a). Phase-contrast images showed cell morphological changes in a DDM dose-dependent manner (Figure 3b). Apoptotic cellular shrinkage, cell fragmentation, membrane blebbing, and detachment traits were observed. Similarly, cell numbers were decreased concomitantly with increasing DDM concentrations.

### 3.4. DDM Release

In favour of determining the pH-dependent drug-releasing properties, the drug release behavior in vitro was studied by using UV–vis in pH originally; 3, 7 and 9 phosphate buffer solutions (PBS) containing DMSO 0.1% to simulate the neutral environment of normal cells and acidic conditions in cancer cells. As shown in Figure 3c, in pH 3, there is a more than 30% release. In sharp contrast, about less than 7% and 0.5% DDM was released at pH 7 and 9, respectively in 24 h, due to protonation and solubility of DDM in acidic environments.

### 3.5. Selective Delivery and Cellular Uptake of DDM in BC Cells

We have confirmed the selective delivery capability of DDM (at IC_50_ doses) into cancer cells by investigating HA, CD44, and FR-α expression levels. As indicated in Figure 3d,e, normal cells (MCF-10A) treated with DDM showed non-significant differences in these levels when compared with the untreated normal cells, indicating that DDM is non-selective to normal cells. As expected, higher HA, CD44, and FR-α expression levels (11-, 4.4-, and 5.3-fold, respectively) were observed in DDM-free MCF-7 cells, and 5.5-, 4.4-, and 5.3-fold higher levels, respectively, in DDM-free MDA-MB-231 cells when compared with DDM-free normal cells. After DDM treatment, the expression levels of HA, CD44, and FR-α were markedly reduced in MCF-7 cells + DDM by 64.6%, 54.6%, and 55.9%, respectively, and in MDA-MB-231 cells + DDM by 43.7%, 53.8%, and 60.4%, respectively, when compared with DDM-free cancer cells. These findings suggested that HA and FA conjugated to DDM is effective surface ligands in targeting the overexpressed CD44 and FR-α receptors on BC cell membranes.

The cellular uptake and localization of DDM in MCF-10A (normal cells), MCF-7, and MDA-MB-231 cells were quantified using AAS (Figure 3f). Quantitative data indicated a higher uptake of DDM by MCF-7 and MDA-MB-231 cells (17.7- and 17.4- fold, respectively) than normal cells, suggesting an increased DDM selective uptake into BC compared to normal cells.

To visualize the internalized nanoparticles and assess their distribution in relation to subcellular compartments, we performed a TEM analysis. In TEM images, numerous high electron density-staining nanoparticles were observed inside the cells treated with DDM, while not observed in cells that were not exposed to DDM (Figure 3g). Normal MCF-10A cells treated with DDM showed a much weaker uptake of nanoparticles than the cancer MCF-7 and MDA-MB-231 cells. In contrast, the uptake nanoparticles from cancer cells treated with DDM were the strongest with significant difference (*p* < 0.001). Because the size primarily influences the composite nanoparticles uptake, the uptake of DDM composite particles was calculated based on the intracellular DDM concentration. When these values are considered against the estimated number of composites introduced to the MCF-7 and MDA-MB-231 cells, ~56 and 81% of nanoparticles, respectively, from IC_50_ dose are more efficiently internalized than normal cells (Figure 3f,g).

### 3.6. DDM Inhibited Tumorigenesis and Enhanced Radiosensitivity of Human Breast Cancer Cells

Next, we performed multi-MTT assays to determine the radio-sensitizing (RS) ability of DDM on BCCs exposed to 3Gy-SDR or 6Gy-FDR following 24, 48 and 72 h (Figure 4a–c). Treatment with IC_50_ dose DDM only significantly decreased the survival of MCF-7 cells to 59.14, 55.12 and 47.83%, and in MDA-MB-231 cells by 61.85, 55.34 and 48.57% at 24, 48 and 72, respectively, as compared to the untreated cells. In the cells exposed to 3Gy-SDR or 6Gy-FDR only, there is no significant difference between the cells survival rate compared to the untreated cells, while a significant increase was observed in MCF-7 cells by 1.64, 1.81 and 2.1 folds and in MDA-MB-231 cells by 1.6, 1.8 and 1.9 folds at 24, 48 and 72, respectively, compared to DDM group. Moreover, we assessed the influence of DDM as radio-sensitizer on the cell’s survival. As shown in Figure 4b,c, DDM + 3Gy-SDR induced a statistically significant reduction in MCF-7 cells survival to 64.6, 48.4 and 22.6%, and in MDA-MB-231 cells to 62.2, 41.2 and 28.4% at 24, 48 and 72, respectively, compared to DDM group. While, in DDM + 6Gy-FDR group set, a highly significant reduction in MCF-7 cells survival to 31.9, 17.2 and 19.3% and in MDA-MB-231 cells to 22.6, 23.9 and 20.4% at 24, 48 and 72, respectively, compared to DDM group. Moreover, as shown in DDM + 3Gy-SDR group, a significant reduction in MCF-7 cells survival when compared with 3Gy-SDR group only to 39.2, 26.7 and 10.8%, and in MDA-MB-231 cells to 38.5, 22.8 and 14.6% at 24, 48 and 72, respectively. In the DDM + 6Gy-FDR group, a highly significant diminution in MCF-7 cells survival was found when compared with 6Gy-FDR only to 19.8, 9.8 and 9.3%, and in MDA-MB-231 cells to 13.9, 14.4 and 10.8% at 24, 48 and 72, respectively. Additionally, DDM + 6Gy-FDR markedly diminished the survival rate in MCF-7 cells to 49.5 and 35.7% at 24 and 48 h, respectively, and in MDA-MB-231 cells to 36.3% at 24 h compared with DDM + 3Gy-SDR (Figure 4b,c). According to our results, generally, the MDA-MB-231 cells demonstrated a greater sensitivity towards the radio-sensitizer DDM than MCF-7 cells, which demonstrated earlier sensitivity to the radiotherapy, whereas MDA-MB-231 revealed later sensitivity (Figure 4d,e). (Figure 4f,g) indicates the results of dose modifying factor (DMF) for all treatments at 24, 48 and 72 h. According to this figure, the dose–response rate in both the 3Gy-SDR and 6Gy-FDR groups combined with DDM was significantly higher than the single treatments, while the DDM + 6Gy-FDR group was also higher than DDM + 3Gy-SDR. These results indicated that DDM induced radio-sensitizing-modified effect with 6Gy-FDR higher than the single treatments and DDM + 3Gy-SDR group. Therefore, the 6Gy-FDR at 24 h was selected for further analysis.

### 3.7. DDM and/or RT Induced Cell Cycle Arrest and Apoptosis in Human Breast Cancer Cells

To investigate the mechanism behind the anticancer activity of IC_50_ doses DDM and increased sensitivity to 6Gy-FDR (RT) in breast cancer cells, we analyzed cell cycle distribution and cell apoptosis by flow-cytometry. As shown in Figure 5a–c, the majority of control cells were blocked in the G1 phase before treatments. However, treatment with DDM induced a marked increase in the proportion of cells in the G2/M and sub G1 phases (apoptosis) compared with the control or RT alone. Additionally, we found the majority of RT-treated MCF-7 cells were blocked in the S phase, and the MDA-MB-231 cells were blocked in the G1 phase compared with the control. When compared to the control or RT alone, combination treatment induced a higher proportion of cells in the G2 and sub G1 phases, and simultaneously a decrease in the percentage of cells in the G1 and the S phase. Furthermore, combination treatment induced a marked proportion of cells in the sub G1 phase compared with DDM only. Cell apoptosis is one of the important determinants of anticancer and radiosensitivity. As shown in flow-based images of cell apoptosis (Figure 5d–f), the percentage of apoptotic cells including early, late and necrotic cells for the DDM or RT groups was increased after 24 h treatment without a significant change for the RT group in MCF-7 cells compared with the control, and apoptotic cells were significantly increased after treatment with DDM + RT compared to control and cells treated with or DDM or RT only. These results suggest that combinatorial treatment of DDM + RT has a potential synergistic effect on the regulation of cell cycle arrest and apoptosis in BCC.

### 3.8. DDM Was a Dual-Targeting for Glycolysis and OXPHOS Pathways and Improved Radio-Sensitivity of Breast Cancer Cells

To verify the mechanism of DDM inhibitory effect on glycolysis and OXPHOS, we tested the expression of glucose, hexokinase, lactate, mRNA-PKM2 (mPKM2), and protein-PKM2 (pPKM2) for glycolytic metabolic pathway as well as HIF-1α, PDK1, PDH, SIRT1, and SIRT3 for mitochondrial-dependent metabolic pathway, after treatment with DDM and/or RT for 24 h in breast cancer cells (Figure 6). The results showed that DDM treatment down-regulated significantly the levels of glucose, hexokinase, lactate, mPKM2, and pPKM2 compared to control or RT only in MCF-7 (Figure 6a,c) and MDA-MB-231 (Figure 6b,c) cells. While the results showed that breast cancer cells treated with RT alone, did not display a significant effect on the glycolytic pathway compared to the control group. To understand the potential mechanisms of DDM-mediated radio-sensitization, we examined the effects of DDM with RT on MCF-7 (Figure 6a,c) and MDA-MB-231 (Figure 6b,c) cells. In both cells, treatment with DDM + RT powerfully down-regulated the levels of glucose, hexokinase, lactate, mPKM2, and pPKM2 than all treatments, except levels of hexokinase, lactate, and pPKM2 in MDA-MB-231 cells as compared with DDM only.

We observed that traditional anticancer drug treatment killed cancer cells, but that some cells survived through ATP induction by OXPHOS activation, and that the surviving cancer cells later acquired drug aggressive growth and resistance. Therefore, the anticancer effect and radio-sensitization of primary therapeutic drugs may be potentiated with OXPHOS inhibitors by DDM. To test this hypothesis, we used breast cancer cell lines (MCF-7 and MDA-MB-231), and we analyzed the expression of HIF-1α, PDK1, PDH, SIRT1, SIRT3, and ROS for mitochondrial metabolic pathway (Figure 6). After 24 h incubation, treatment with DDM exhibited a significant decrease in the levels of HIF-1α, PDK1, PDH, SIRT1, ROS and an increase in SIRT3 level in MCF-7 (Figure 6d–g. The uncropped Western blots have been shown in Appendix A.) and in MDA-MB-231 cells (Figure 6e–g) compared with control or RT only. The results showed that RT alone did not display a significant effect on mitochondrial metabolic pathway compared to the control group in both cell lines. The combination of DDM + RT exerted a significant decrease in the levels of HIF-1α, PDK1, PDH, SIRT1, and ROS, and an elevation in SIRT3 level in MCF-7 (Figure 6d–g) and in MDA-MB-231 cells (Figure 6e–g) compared with all treatments except for PDK1, PDH, SIRT1, SIRT3, and ROS in both cells, indicating that DDM has a dual potency as an anti-proliferative and radio-sensitization.

### 3.9. DDM and/or Radiotherapy Regulated Metabolic Glycolysis and OXPHOS through a Novel PI3K/AKT/mTOR/P53/NF-κB/VEGF Signaling Pathway in Breast Cancer Cells

To further explore the underlying mechanism of metabolic glycolytic and mitochondrial pathways, we investigated a series of relative key proteins involved in the Warburg effect and cell energy metabolism regulation including PI3K, AKT, mTOR, P53, NF-κB, and VEGF. DDM significantly down-regulated the levels of PI3K, AKT, mTOR, NF-κB, and VEGF along with induced up-regulation in the P53 level in MCF-7 (Figure 7a–c) and MDA-MB-231 cells (Figure 7d–f) compared with control or RT only except for VEGF level in MCF-7-treated with RT only. The results also demonstrated that treatment with RT alone induced a moderate effect on the regulated pathway of the glycolysis and OXOHOS by reducing PI3K, AKT, mTOR, NF-κB, and VEGF levels in parallel with the up-regulated the P53 level in both cells except for VEGF in MDA-MB-231 compared with the untreated cells. Furthermore, in an attempt to elucidate the mechanism of DDM in the enhancement of radiation effect on the regulated pathway of metabolic glycolysis and OXPHOS, the result revealed that upon radiation treatment following DDM a significant down-regulation in the level of PI3K, AKT, mTOR, NF-κB, and VEGF coupled with an up-regulation in the P53 level in MCF-7 (Figure 7a–c) and MDA-MB-231 cells (Figure 7d–f) compared with all treatments except for NF-κB and VEGF when compared with DDM only, suggesting that DDM augmented the effect of radiation by inhibiting glycolysis and OXPHOS through the modulation of the PI3K, AKT, mTOR, P53, NF-κB, and VEGF signaling.

## 4. Discussion

Our findings are in agreement in part with the earlier observations reported by Warburg termed the Warburg effect. The Warburg premise states that cancer cells switch from mitochondrial energy to glycolysis for ATP production, which is blocked by 2DG via HK targeting [29]. However, it has not been sufficient to discontinue cancer progression. As reported by a previous study, cancer stem cells predominantly pursue their metabolism via OXPHOS phenotype rather than glycolysis [30]. Furthermore, DCA promotes pyruvate influx into mitochondria via targeted pyruvate dehydrogenase kinases [29]. However, glycolytic and mitochondrial bioenergetics suppression by multiple cancer development mediators currently induced by DDM could function as potential therapeutic options against BC as demonstrated by our findings. The innovative formula of DDM core–shell exhibited several characteristics. Firstly, it consisted of an MgO core for enhancing the potency of radio-sensitizer namely, 2DG and DCA with a pH-sensitive degradable polymer shell for the controlled release of a targeted-drug-DDM in response to acidic tumor conditions. Recently, MgO-NPs as potential candidates for drug delivery, anticancer, magnetic resonance imaging, and hyperthermia systems have been studied [11]. Secondly, the possible incorporation of contrast agents such as MgO into nanoparticles could help in tracking the drug uptake by TEM imaging. Thirdly, it allows for the simultaneous delivery of chemotherapeutic reagents (2DG and DA) and radio-sensitizers (e.g., MgO). Finally, HA and FA molecules on the DDM surface help target overexpressed CD44 and FR-α receptors in BC cells, thereby facilitating site-specific targeted therapy.

Diffraction characteristics are displayed inside 2θ (degree) as 24.20, 32.60, 37.50, 58.93, 62.03, and, 72.18 where some peaks represent the Bragg’s appearances (001), (111), (20), (220), (311), and (222) extensions in that position sequentially, which can be recorded to the levels of cubic MgO (JCPDS 75-0447) [31]. This suggests that the MgO-NPs (core structure) were regularly crystal in fact and produced the FCC crystalline configuration. It must be stated that the amorphous peak at 17.35 (*) was due to outer organic shells DDM. The presence of O, C, Cl, and Mg is confirmed where the existence of Mg, and O atoms is confirmed for the core MgO-NPs by EDX. Moreover, the presence of O, C, and Cl is attributed to the DA, 2DG, HA, and FA multi-shell structures in the synthesized sample [12,32]. It is evident from these images that the elements Mg, C, Cl, and O exist, which agreed with preceding EDX results. Further, those elements are homogeneously distributed. From the images, we can conclude that both Mg (blue color), and O (pink color) atoms are located in the same places, which confirms the core structure. The surface behavior reveals dark layers represent the outer shells (HA and FA) with remarkable smooth agglomerates can be observed due to the occupation of a large quantity of layers at the grain boundary which could control the grain growth [33]. Moreover, the bright particles represented the MgO-NPs core which confirms the promising core–shell structure. The synthesized composite possesses semi-spherical structure with diameter sizes ranging from 165.21–88.94 nm with an average size of 99.87 nm. It must be noted that the condensed particles were attributed to the core MgO-NPs (yellow circles), while the faint layers corresponded to the shell layers DDM which are entirely validated by color in mapping/SEM images. For the present, nanocomposite the characteristic vibration peak at 680 cm^−1^ was assigned to the stretching mode of MgO (in the core) and was in a good agreement with the literature [34,35,36].

In FTIR results, the characteristic IR absorption peaks at 1608, 1776 and 1501 cm^−1^ are observed in the spectrum which is assigned to FA and due to N–H bending vibration of CONH group, C=O amide stretching of the α-carboxyl group, and the absorption band of the phenyl ring, respectively [37]. The presence of a band at 3696 cm^−1^ is attributed to OH and NH stretching regions. The band at 2965 cm^−1^ can be attributed to the stretching vibration of C–H in HA. The band at about 1776 cm^−1^ corresponds to the amide carbonyl and the band at 1440 cm^−1^ can be attributed to the stretching of COO−, which refers to the acid group of molecule HA. The absorption band at 1089 cm^−1^ is attributed to the linkage stretching of C–OH in HA [38]. DCA normally exhibits a peak at around 1836 cm^−1^ assigned to the C=O stretching vibration for the COCl group. On the other hand, the spectrum of DCA has a peak at 1776 cm^−1^ assigned to the C=O of the COOH group and peaks at around 1341, and 1272 cm^−1^ assigned to the O–H and C–O, respectively. In addition to these, the 1608 cm^−1^ peak assigned to the COO− anti-symmetric stretching vibration were also observed in the spectrum of DCA [39]. The presence of 2DG in the synthesized nanocomposite is indicated by peaks at 1272 cm^−1^ for O–H blending of 2DG and 1089, and 930 cm^−1^ for C–O stretching of 2DG [40]. Finally, the peak at 680 cm^−1^ was assigned to the stretching mode of MgO in the MgO-NPs core.

After the literature comparison achieved between the FTIR data of bare FA [41], HA [42], DCA [43], 2DG [44], and bare MgO-NPs [45], it must be noted that the connection type between the outer organic shells was by covalent bond due to the presence of new peaks as described before, which were not present in bare FA, HA, DCA, 2DG, and bare MgO-NPs which indicated the conjugation behavior between layers and indicated by a covalent bond, as described in the recent publications [46,47,48]. In our FTIR results, the conjugation behavior was detected as new peaks formed (strong covalent bond) or a minor shifting in the bare peaks (weak physical bond) [49,50].

In this study, DDM was improved by applying a more effective pegylation strategy using a functionalized core–shell nanoparticle, in contrast to nanoparticles used in previous studies [51]. This approach increased stability in bio-relevant media and improved functionalization with HA and FA-targeting ligands. Our DDM had an average hydrodynamic diameter size of approximately 330 nm over 6 days, which was appropriate for tumor tissue accumulation and adhering to EPR principles [5]. The zeta potential value of DDM demonstrated good stability, perhaps because electrostatic repulsion between particles was dominated by van der Waals attraction forces, ensuring stability in solution. The DDM did not aggregate and was adequately stable in bio-relevant media, suggesting a good retention of optimum size characteristics in the blood circulation. This enhanced stability may be attributed to the novel core–shell design which could reduce interactions between nanoparticles and bio-relevant media constituents, e.g., proteins. Protein binding to nanoparticle surfaces in vivo could result in rapid uptake by the reticuloendothelial system and removal of nanoparticles from the general circulation, reducing or eliminating accumulation at tumour sites [5,52].

We evaluated DDM cytotoxicity in BC cells using the MTT assay, and showed the inhibitory capability of DDM on BC cell proliferation in a dose-dependent manner. Previous studies reported that the suppression effect of 2-DG or DCA by itself was limited since the cytotoxic dose did not significantly increase the inhibition ratio [53,54]. Additionally, cytotoxicity curves indicated that MDA-MB-231 cells were more affected by DDM than MCF-7 cells. This was probably because DDM was targeted more to cells which were characterized by more mesenchymal (drug resistance), and relying on glycolysis for ATP production (Warburg effect) under both normoxic and hypoxic conditions. Moreover, MDA-MB-231 cells grew considerably faster (~1.5×) than MCF-7 cells [55].

Cell surface-specific markers have been extensively used for cancer-targeted therapies [56]. Among these, CD44 and FA-α receptors have been studied for selective nanoparticle drug delivery to BC sites. According to previous research, nanoparticle size primarily influences composite nanoparticle uptake [57]. In this study, selective cell uptake was used to calculate and image DDM in BC cells. Cells treated with DDM showed much stronger fluorescent signal uptake than normal cells, suggesting synergic effects from HA/FA-mediated endocytosis. Hence, dual-receptor-mediated synergic internalization significantly improved drug system selectivity and targeted efficiency to BC cells. Additionally, DDM cellular uptake was significantly decreased in normal cells, suggesting the CD44 and FR-α dependent uptake, which facilitated the enhanced intracellular drug concentration toward both CD44 and FR-α receptors which overexpressing in cancer cells. Our findings agreed with a previous study showing that improved cytotoxic effects of drugs on cancer cells were attributed to drug release under acidic conditions [58]. It is worth mentioning that the encapsulation system of the nanoparticles maintains their structural integrity at 1.5 pH and the contents can only be released slowly at pH > 5, which suggests the stability of the encapsulation system under the conditions of a simulated stomach and release the contents slowly under the conditions of intestinal tract [59]. These promising results indicate that DDM design can be used via different routes of administration with pronounced stability under a wide range of pH (3–9 pH) as revealed in our data. Accordingly, the novel nano-core–shell targeted DDM formulation possesses anticancer characteristic with enhanced targeting towards breast cancer cells and apparent stability with maintained patient compliance later on. Worthwhile, the simultaneous release of the active components (2DG, DCA, and MgO) takes place within the tumor microenvironment (at pH 3 and upwards to approximately pH 5) after the active targeting via the outer shell annex (HA and FA). The drug release kinetics of DDM in acidic (pH 4) conditions demonstrated the feasibility of the DDM formulation in cytoplasmic drug release after CD44 and FR- receptor-mediated endocytosis in BC cells. Thus, our qualitative and quantitative data revealed that the DDM system is specifically and selectively bound to the overexpressed FR- and CD44 receptors in cancer cells via CD44/FR- receptor-mediated endocytosis and the responsive release under acidic conditions. These findings, associated with the mechanism of action, indicated that DDM is a promising platform for BC therapy.

Although tumor cell radio-sensitization is considered a promising strategy to combat cancer, it is equally important to reduce RT mediated toxic effects in healthy tissues surrounding tumors. Thus, using drug/RT combinations could permit increased tumor control even for radio-resistant triple-negative BC tumors. The main impact of RT is reactive oxygen species (ROS) generation which damages DNA and causes cell death via apoptosis. Importantly, ROS production also contributes to malignancy. Nevertheless, various tumors acquire a radio-resistance that is accountable for the failure of RT and recurrence of tumor or metastasis, due to HA overexpression and low oxygen levels (hypoxia) in the tumor-microenvironment [60,61]. In spite of the effectiveness of DDM alone at all intervals of time over the group of cells treated by RT only in the reduction of cells survival, inducing radio-sensitization with ROS reduction by DDM may represent a plan to overcome tumor radio-resistance and improve the therapeutic outcomes. Moreover, our combined chemo-radiotherapy at 6Gy-FDR showed increased radio-sensitivity in BC cells, especially MDA-MB-231 cells, which is in agreement with previous studies [62,63]. Our findings revealed that when the RT dose was increased to 6Gy-FDR + DDM, cell death and DMF rates were significantly higher than other treatments. These findings indicated the effectiveness of DDM + 6Gy-FDR toward cell death and induced modified effects. When compared with 3Gy, the energy transferred to cells at 6Gy-FDR was sufficient to cause cell death, in agreement with Ebrahimi et al. [64]. It appeared that 6Gy-FDR + DDM exerted significant effects on cell survival reduction and rapidly growing cells (MDA-MB-231) when compared with MCF-7 cells. Analysis of the radio-modifying-effects of DDM in BC cells showed that the DDM administration time with respect to RT was important in determining effects. Stimulation is generally higher after exposure to FDR for 24 h. Because Jalil et al. [65] found that the majority of induced damage response pathways function optimally for a few hours after RT, therefore the 6Gy-FDR (RT) was chosen for further investigation.

Cell cycle regulation and cell death signaling pathways have important roles in cancer development; therefore, they serve as potential cancer therapeutic targets [66]. Indeed, compounds that induce cell cycle arrest and apoptosis may function as radio-sensitizing reagents and valuable strategies for cancer drug discovery [67,68]. We determined the effects of DDM and/or RT on cell cycle progression and apoptosis in BC cells and showed the cell cycle was arrested at the G2/M phase, and apoptosis induced by DDM treatment in BC cells. As confirmed by previous research, the radio-sensitizing effects of anticancer drugs are due to cell cycle alterations and apoptosis progression. 2-DG is one such anticancer drug and was shown to enhance radio-sensitivity by blocking cells in the G2/M phase of the cell cycle and enhancing apoptosis [69]. Our results indicate that DDM treatment before RT increased radio-sensitivity in BC cells via G2/M arrest. More interestingly, we observed an accumulation in the S phase for MCF-7 cells and G1 phase for MDA-MB-231 cells, with no significant differences in Sub G1 when exposed to radiation only. This may be due to that AKT decrease that leads to G1/S arrest in various cancer cells when exposed to RT, as reported by Williams and Schumacher [70]. A radiation-induced cell cycle checkpoint is believed to provide cells with additional time to repair DNA damage before further engagement with the cell cycle. Delays at the G1 phase after exposure to radiation is characteristic for cells expressing a wild-type P53 pathway [71]. Therefore, our results showed that the simultaneous exposure of BC cells to DDM, with or without RT, accelerated the cell cycle through the G1/S and arrested it in the G2/M phase. Apoptosis is a major cancer suppression mechanism and is characterized by morphological and ultra-structural changes associated with biochemical processes [72]. DDM incorporates promising complex agents, including 2DG, DCA, and MgO which induce apoptosis and act against BC cells [27,29,73], although they do not show strong cell death when used separately. Our annexin-V data indicated that DDM induced apoptosis in BC cells and showed potential anti-proliferative and apoptogenic effects when agents were combined. Additionally, low apoptosis levels mediated by RT in MDA-MB-231 cells may have been related to induced resistance in these cells; this observation agreed with previous work confirming that radio-sensitivity in MDA-MB-231 cells was significantly lower than MCF-7 cells [74]. While FDR following DDM improved cell death when compared with RT, perhaps DDM induced CD44 suppression and FA-receptor overexpression which contributed to poor radio-responses and chemo-resistance. However, our results indicated that DDM followed by RT strongly enhanced cell death in BC cells.

Multiple genes are involved in the Warburg effect (aerobic glycolysis) and mitochondrial metabolism controlled processes. Thus, modifying the expression of just one gene may be insufficient to suppress tumors, potentially facilitating cancer progression and drug resistance [75]. Glucose, hexokinase [76], PKM2 [69], lactate [77], PDH, PDK1, SIRT1, and SIRT3 [76], are essential glycolytic and mitochondrial metabolism enzymes controlling tumor progression or regression. Previous studies have reported that blocked glycolytic and mitochondrial metabolic enzymes constituted major targets in preventing cancer growth [78,79], consistently with DDM effects in this study. HK2 is the isoform expressed in cancer cells and regulates the first step in glycolysis. It is regulated by P53 and HIF-1 and is reported as the target of 2DG. Another important kinase in glycolysis is pyruvate kinase M2; it is phosphorylated by PI3K/AKT, thereby promoting the Warburg effect and tumorigenesis [80]. Another kinase is pyruvate dehydrogenase kinase-1 (PDK1), which is reportedly a target of DCA. Rather than converting pyruvate to lactate, it regulates the transformation of pyruvate to mitochondria via pyruvate dehydrogenase (PDH) [81]. Two mitochondrial sirtuins are SIRT1 and SIRT3. The latter is the major mitochondrial sirtuin which promotes ATP production by regulating tricarboxylic acid (TCA) cycle enzymes [82].

Previous studies have reported that blocked glycolytic and mitochondrial metabolism enzymes constitute major targets in preventing cancer [78,79], consistent with the DDM effects seen in this study. Our results suggested that DDM inhibited the metabolic reprogramming of BC, perhaps due to the major targeting mediators’ inhibition in aerobic glycolysis and mitochondrial metabolism. To control the glycolysis pathway, DDM induces a significant decrease in glucose uptake, hexokinase, PKM2, and lactate levels, indicating potential effective of DDM in Warburg effect blockage, maybe by dephosphorylation of HK and PKM2 through PI3K/AKT inhibition, thereby preventing the Warburg effect and cancer progress, while, for controlling of OXPHOS pathway, it induced a down-regulation of ROS production, PDH phosphorylation, and PDK1 and SIRT1 activity, also an up-regulation of SIRT3, activated several proteins in the apoptotic pathway as annexin-V and p53, and then induced cell death, indicating a greater effective potency than anticancer drugs which targeted a single pathway [83,84]. Previous studies reported that DCA plus a SIRT inhibitor exerted anticancer activity via P53 acetylation in MCF-7 cells [76,85]. Our study suggested that SIRT1, SIRT3 and ROS, as OXPHOS regulators at the mitochondria, SIRT1 overexpression, and SIRT3 down-expression are associated with glycolysis and proliferation through dependent-P53 in MCF-7 and MDA-MB-231 cells.

Because cancer cells use the Warburg effect and OXPHOS for energy production, the conversion of pyruvate into acetyl CoA is irreversible and future therapies targeting dual cellular metabolism (glycolysis and OXPHOS) should be designed [22,84]. These results demonstrate that although the anticancer efficacy of drugs is ROS dependent [85], the DDM/or RT approach is not. This may be due to P53 stimulation, which induces the transcription of the TIGAR gene, which lowers the fructose-2,6-bisphosphate (F-6-bP) and thus decreases glycolysis, mitochondrial ATP production, and overall levels of ROS, suggesting that DDM/or RT is a promising anticancer strategy through the targeting of OXPHOS in BC cells. In our study, poor radiation responses observed in BC cells may have been due to the tumors use of the cellular environment through a hypoxic cellular environment, enhanced glycolysis or OXPHOS and ROS elevation, which results in the radio-resistance in agreement with Pajak et al. [86]. Multiple studies have demonstrated that ROS may have a role as a Warburg effect stimulant via HIF-1α in response to hypoxia concordantly with our research [84,87]. In our study, changes in gene and cellular levels indicated that hexokinase, PKM2, lactate, PDH, PDK1, SIRT1, and SIRT3 promoted the Warburg effect and OXPHOS mechanisms in MCF-7 and MDA-MB-231 cells. We observed that DDM added prior to RT enhanced radiation-induced cell death via modulated metabolic reprogramming. This observation agreed with previous research demonstrating that 2-DG added prior to or immediately after IR, enhanced radiation-induced cell death by modifying energy-dependent cellular processes, e.g., DNA damage repair, cell cycle checkpoints, and apoptosis [27,87]. We suggest that DDM selectively blocks both glycolysis and OXPHOS pathways and enhances radio-sensitivity in BC cells via the down-regulation of mediators in these processes through PI3K/AKT/mTOR/P53NF-κB/VEGF signaling pathways.

## 5. Conclusions

In this study, pegylated DDM was functionalized with HA and FA. These core–shell nanoparticles exhibited controlled 2DG, DCA, and MgO, even in bio-relevant media (cell culture medium), which was accelerated at acidic pH. The HA and FA-functionalized DDM only or as a radio-sensitizer before RT increased cell cycle arrest, apoptosis, and cytotoxicity against BC cells. Moreover, DDM exhibited increased uptake and increased cytotoxicity against MDA-MB 231 cells when compared with MCF-7 cells. Thus, glycolysis/OXPHOS inhibition by DDM and FDR treatment may induce cancer metabolic reprogramming via a novel PI3K/AKT/mTOR/P53NF-κB/VEGF pathway in BC cells. Therefore, the dual targeting of glycolysis/OXPHOS pathways is suggested as a promising antitumor strategy.

## Figures and Tables

**Figure 1 cancers-13-05571-f001:**
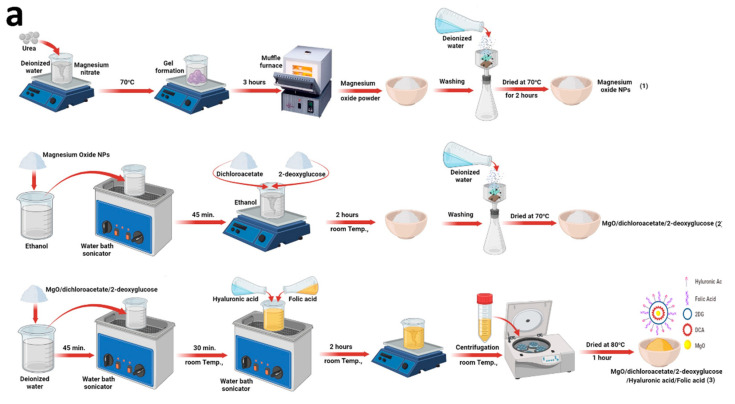
Characteristics of DDM. (**a**) Diagram of DDM synthesis. (**b**) The crystallinity behavior of MgO-NPs, and amorphous shape of organic shells in the synthesized DDM by XRD analysis. (**c**) EDX spectra of the synthesized DDM. (**d**) Elemental mapping images of the synthesized DDM.

**Figure 2 cancers-13-05571-f002:**
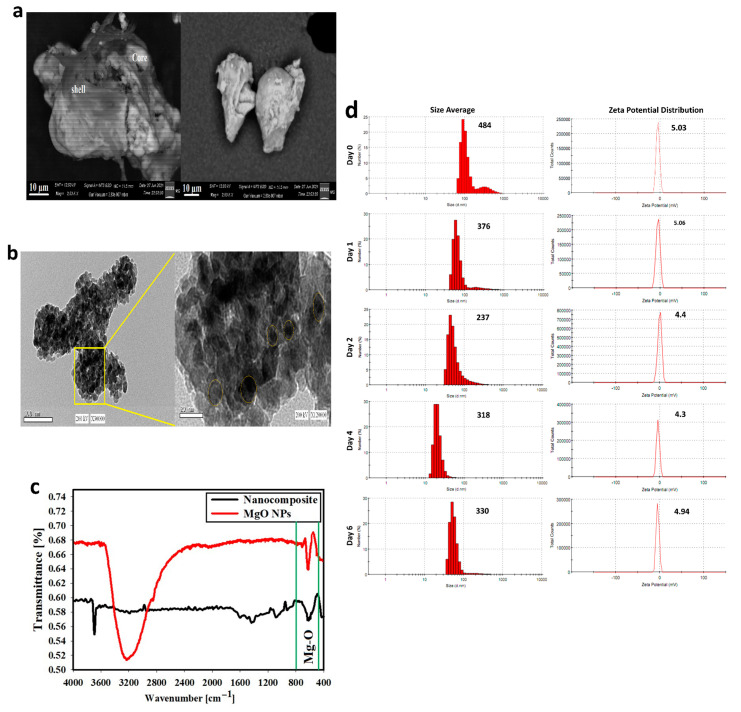
Structural characteristics of DDM. (**a**) SEM images of the synthesized core-shell DDM sample at different magnifications. (**b**) HRTEM images of the synthesized core-shell DDM at different magnifications. (**c**) FTIR analysis of the synthesized core-shell DDM. (**d**) Stability of DDM in PBS containing 10% FBS indicated by DLS and zeta potential analyses.

**Figure 3 cancers-13-05571-f003:**
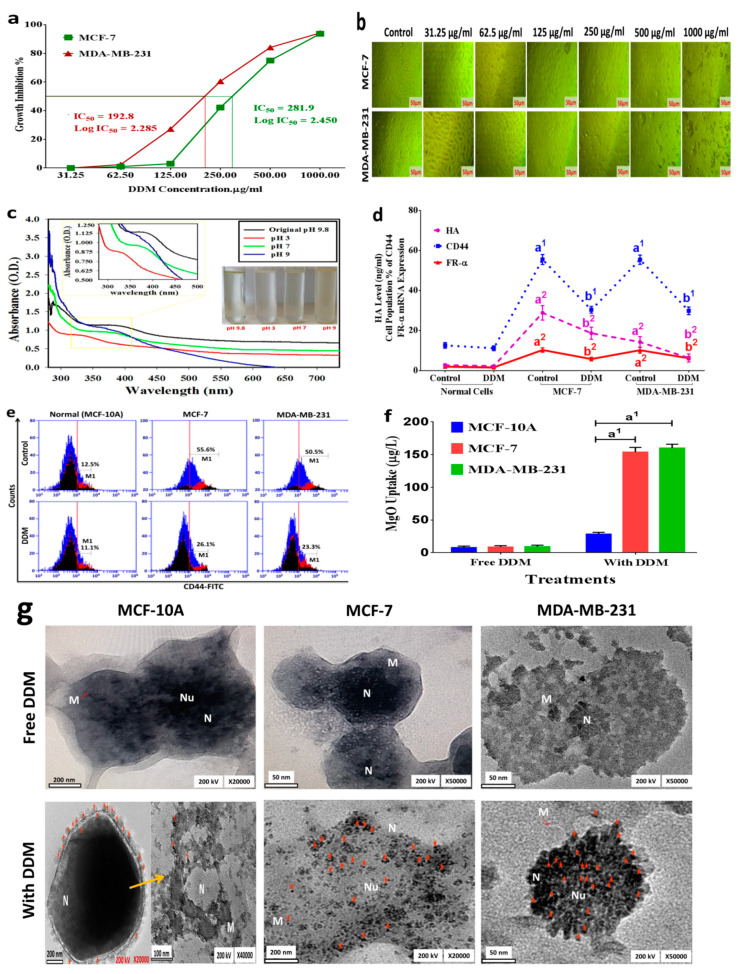
Inhibition of breast cancer (BC) cell proliferation, release, enhanced selectivity, and cellular uptake by DDM. (**a**) MTT assay of BC cells treated with DDM at different concentrations. Values are represented as the mean ± standard error of the mean (SEM) of triplicate samples from two independent experiments. (**b**) Phase-contrast images of MCF-7 and MDA-MB-231 cells were captured by the inverted light microscope. (**c**) The release of DDM at different pH; originally 3, 7, and 9 phosphate buffer solutions (PBS) containing DMSO 0.1% estimated using UV-vis spectrometry. (**d**) Representative histogram of the selective HA, CD44, and FR-α expression in BC cells. (**e**) Flow-cytometric images showing a single parameter histogram for CD44 in different groups, where the M1 denotes the cell population stained with the CD44 antibody. (**f**) Atomic absorption spectrophotometry (AAS) intracellular MgO uptake data in MCF-10A, MCF-7, and MDA-MB-231 cells after 24 h incubation with DDM at IC_50_ doses. (**g**) Transmission electron microscopy (TEM) images of MCF-10A, MCF-7, and MDA-MB-231 DDM uptake after 24 h incubation with DDM at IC_50_ doses. Data are represented as the mean ± standard error of the mean (SEM) (*n* = 3). ^a1^
*p <* 0.001, ^a2^
*p <* 0.01 vs. untreated normal cells; ^b1^
*p <* 0.001, ^b2^
*p <* 0.01 vs. untreated cancer cells.

**Figure 4 cancers-13-05571-f004:**
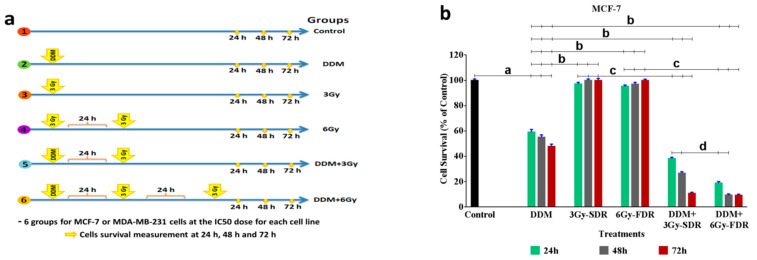
Cell survival and radio-sensitization rates using the multi-MTT assay. (**a**) Experimental schematic outline of the six different treatment groups. Evaluation of cell survival (**b**,**c**), percent inhibition for MCF-7 (**d**), percent inhibition for MDA-MB-231 (**e**), and dose-modifying effect curves (**f**,**g**) in response to DDM, 3Gy-SDR, or 6Gy-FDR, and a combination of both at 24 h, 48 h and 72 h using the multi-MTT assay. Data are the mean ± standard error of the mean (SEM), where control cells are 100% (*n* = 3). ^a^
*p* < 0.01 vs. control; ^b^
*p <* 0.01 vs. DDM group; ^c^
*p* < 0.001 vs. 3Gy-SDR or 6Gy-FDR groups; ^d^
*p* < 0.05 vs. 3Gy-SDR+DDM group.

**Figure 5 cancers-13-05571-f005:**
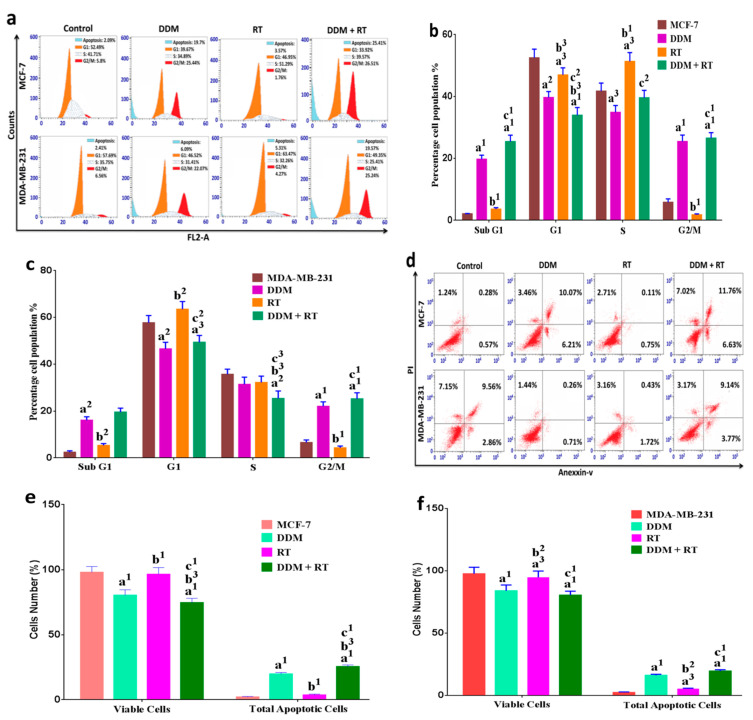
DDM and/or RT induced cell cycle arrest and apoptosis in BC cells. (**a**–**c**) Representative cell cycle images of MCF7 and MDA-MB-321 cells at G2/M arrest after a 24 h treatment with DDM or DDM + RT, and cycle arrest at S phase in MCF-7 and G1 phase in MDA-MB-231 when treated with RT only. (**d**–**f**) A representative dot plot of annexin-V-FITC and PI staining in BC cells treated with DDM or DDM + RT for 24 h showing increased apoptotic cells when compared with controls or RT alone. Slightly increased apoptotic cells are seen in the RT group compared to the control. Data are the mean ± standard error of the mean (SEM). ^a1^
*p <* 0.001, ^a2^
*p <* 0.01, ^a3^
*p <* 0.05 vs. control; ^b1^
*p <* 0.001, ^b2^
*p <* 0.01, ^b3^
*p <* 0.05 vs. DDM group; ^c1^
*p <* 0.001, ^c2^
*p <* 0.01, ^c3^
*p <* 0.05 vs. RT group.

**Figure 6 cancers-13-05571-f006:**
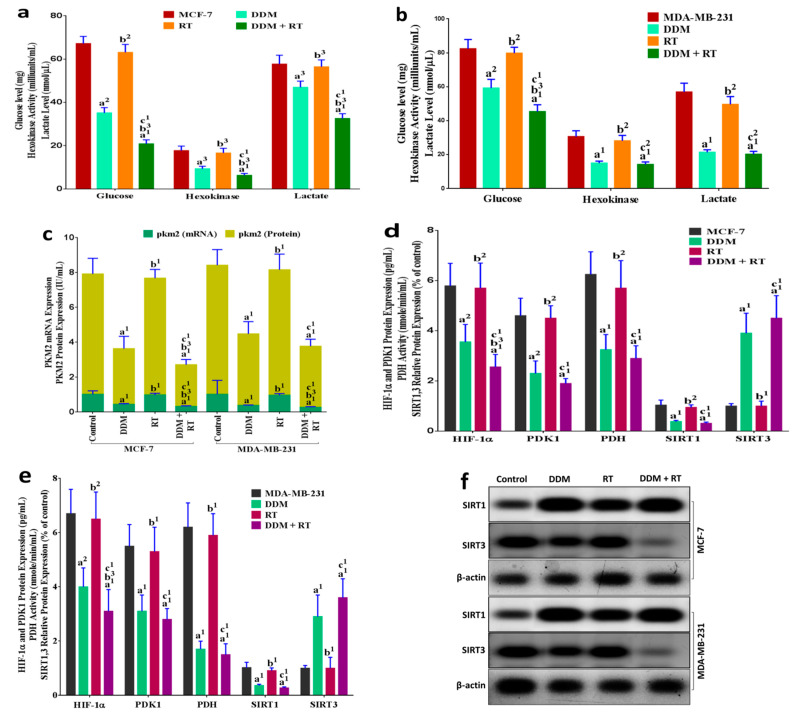
The influence of DDM and/or RT on glycolysis and mitochondrial metabolic pathways in BC cells. (**a**,**b**) Glucose, hexokinase, and lactate levels were analyzed by colorimetric assay in all groups. (**c**) mRNA and protein expression of PKM2 was rescued in all groups. (**d**,**e**) HIF-1α, PDK1, SIRT1, and SIRT3 protein expression and PDH enzymatic activity was rescued in all groups. (**f**) Relative SIRT1, SIRT3, and β-actin protein levels in all groups. (**g**) ROS levels (ELISA) in all groups. All group values are given as the mean ± standard error of the mean (SEM). ^a1^
*p <* 0.001, ^a2^
*p <* 0.01, ^a3^
*p <* 0.05 vs. control; ^b1^
*p <* 0.001, ^b2^
*p <* 0.01, ^b3^
*p <* 0.05 vs. DDM group; ^c1^
*p <* 0.001, ^c2^
*p <* 0.01, ^c3^
*p <* 0.05 vs. RT group.

**Figure 7 cancers-13-05571-f007:**
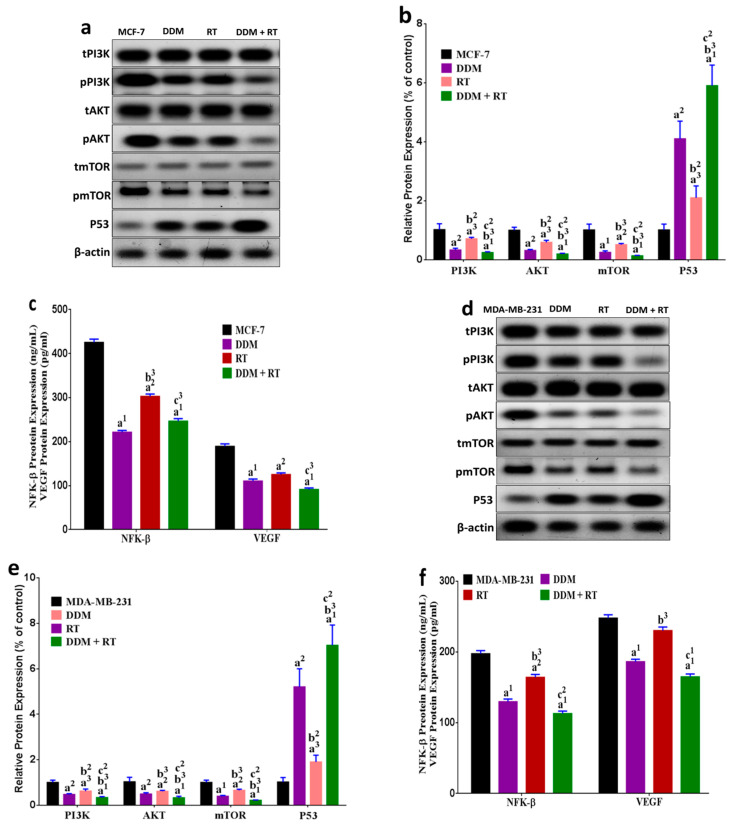
DDM and/or RT modulate key proteins involved in the regulation of glycolysis and OXPHOS pathways in BC cells. PI3K, AKT, mTOR, P53, and β-actin protein levels in MCF-7 cells (**a**,**b**) and MDA-MB-231 cells (**d**,**e**). NF-κB and VEGF levels (ELISA) in both cell groups (**c**,**f**). (**g**) Schematic representing the mode of action of targeted multifunctional core–shell-nano (DDM) and RT. The DDM core–shell is a chemotherapy drug targeted to cancer cell surfaces via cancer cell-specific ligands. DDM binds to surfaces by recognizing specific receptors resulting in DDM internalization via endocytosis. Inside the cell, DDM undergoes endosomal escape leading to cytotoxic drug release. Treatment with DDM alone or as a radio-sensitizer (prior to RT) induces cancer cell death through modulating the metabolic reprogramming displayed by BC cells via a novel PI3K/AKT/mTOR/P53/NF-κB/VEGF pathway. (→) Indicates pathway direction and (T) indicates blocking functions. All group values are presented as the mean ± standard error of the mean (SEM). ^a1^
*p <* 0.001, ^a2^
*p* < 0.01, ^a3^
*p* < 0.05 vs. control; ^b2^
*p* < 0.01, ^b3^
*p* < 0.05 vs. DDM group; ^c1^
*p <* 0.001, ^c2^
*p <* 0.01, ^c3^
*p <* 0.05 vs. RT group.

**Table 1 cancers-13-05571-t001:** The primers’ sequence for quantitative real-time PCR.

Genes	Forward Primers	Reverse Primers
FR-α	5′-CTGGCTGGTGTTGGTAGAACAG-3′	5′-AGGCCCCGAGGACAAGTT-3′
PKM2	5′-GAGGCCTCCTTCAAGTGCTG-3′	5′-CATGGCAAAGTTCACCCGGA-3′
GAPDH	5′-GTCAAGGCTGAGAACGGGAA-3′	5′-AAATGAGCCCCAGCCTTCTC-3′

## Data Availability

All data generated or analyzed during this study are included in this manuscript.

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
