# Peer review of "Dual Hyaluronic Acid and Folic Acid Targeting pH-Sensitive Multifunctional 2DG@DCA@MgO-Nano-Core–Shell-Radiosensitizer for Breast Cancer Therapy"

_cancers, 2021, doi:10.3390/cancers13215571_

Round 1

Reviewer 1 Report

1. Cells without treatment is not a proper control. What is the vehicle used to suspend the nanoparticles? 
2. There is no in vivo data at all. What is the goal of making nanoparticles? Even water can kill cells. 
3. What does M1 stand for in Fig 2e?
4. There is a major problem with data analysis. In Fig 2f, did you plot the mean and SD of MgO uptake from MCF-10a, MCF7, and MDA-MB-231? Same problem in Fig 3d.
5. Fragmentary figures: Fig 4a,4d.

Author Response

13-Oct-2021

Letter to the reviewers of Cancers

Subject: Compliance to reviewers’ comments in regards of the Manuscript # cancers-1406449.

Dear Prof. Dr. Samuel C. Mok

Editor in chief of cancers Journal

It is a great pleasure to have this chance of publishing in your respective journal “Cancers”. Herein, we are trying to respond precisely to all the valuable points suggested by reviewers and hope that it will fulfil the journal requirements and make our research deserve this opportunity to be published through this great platform.

Reviewers

I am happy to report that this version of the manuscript is significantly improved compared with the previous version I read. The relevant parts of manuscript have been updated and improved.

Finally, we would like to thank the reviewers for the constructive comments to improve our manuscript.

Thanks again for your time.

Response to Reviewer 1 Comments

Point 1: Cells without treatment is not a proper control. What is the vehicle used to suspend the nanoparticles?

Response 1: The vehicle DMSO, at the 0.1% concentration used in the experiments. Cells treated with the vehicle were the negative control, and has been added in the manuscript.

Point 2:There is no in vivo data at all. What is the goal of making nanoparticles? Even water can kill cells.

Response 2: This study was designed to promptly address the major cytotoxic profile of the novel fabricated nano-core-shell platform for investigating the efficacy of innovative targeted formulation based on 2DG@DCA@MgO against certain types of breast cancer cell lines; MCF-7 & MDA-MB-231, it is therefore, the findings obtained in the current study will be utilized in expanding our aim to include in vivo study in a future perspective. To the best of our knowledge, in vitro studies are used as a verified preliminary data guiding to study the drug in vivo. Considering that DDM is applied for the first time, we believe that in vitro study is the most proper mean of evaluation before the transfer to in vivo studies. This study covered the major points of the DDM anticancer effects in vitro as compared with normal and untreated cells, indicating that DDM enhanced selectivity towards the breast cancer cells devoid of apparent cytotoxic effects against normal cells (MCF-10A) or vehicle- treated cells. Various previous studies have reported the newly formulated nanoparticles as a promising anticancer drugs through studies based on in vitro data(1- 3). Hopefully, the current findings will be used in a further study conducting in vivo model as per reviewer recommendation.

With reference to the reviewer query about the goal of using nanoparticles in our study, the authors intended to utilize nanoparticles owing to the several advantages they possess superior to the conventional compounds used in normal scale, bearing in mind that these merits enables the augmented targeting against a broad panel of cancers. These advantages including the facilitated penetration into tumor cells enhanced permeability and retention (EPR) effects, thereby increasing cancer therapeutics efficacy with special emphasis when tumor- cell surface markers (CD44 & folate receptors) are targeted.

  • Chan L, et al. Cancer-Targeted Selenium Nanoparticles Sensitize Cancer Cells to Continuous γ Radiation to Achieve Synergetic Chemo-Radiotherapy. Chem Asian J. 2017 Dec 5;12(23):3053-3060. doi: 10.1002/asia.201701227.
  • AthinaAngelopoulou, et al. Folic Acid-Functionalized, Condensed Magnetic Nanoparticles for Targeted Delivery of Doxorubicin to Tumor Cancer Cells Overexpressing the Folate Receptor. ASC omega J 2019, 4, 22214-22227. DOI.org/10.1021/acsomega.9b03594.
  • Markowski A, et al. Evaluation of the In Vitro Cytotoxic Activity of Ursolic Acid PLGA Nanoparticles against Pancreatic Ductal Adenocarcinoma Cell Lines. Materials (Basel). 2021 Aug 29;14(17):4917. doi: 10.3390/ma14174917.

Point 3: What does M1 stand for in Fig 2e?

Response 3: The M1 is the cell population stained with the CD44 antibody, and has been added in the manuscript.

Point 4: There is a major problem with data analysis. In Fig 2f, did you plot the mean and SD of MgO uptake from MCF-10a, MCF7, and MDA-MB-231? Same problem in Fig 3d.

Response 4: Has been clarified in Figure 2f, and 3d in the manuscript

Point 5: Fragmentary figures: Fig 4a,4d.

Response 5: Has been corrected in the manuscript

Reviewer 2 Report

In the present paper, authores developed CD44 and folate receptor-targeting multi-func-38 tional dual drug-loaded nanoparticles (DDM). These contained a hyaluronic acid (HA) and folic 39 acid (FA) conjugated to a dichloroacetate (DCA) shell linked to a 2-deoxy glucose (2DG) and mag-40 nesium oxide (MgO) core to enhance localized chemo-radiotherapy for effective BC treatment. However well structurated and with promisor results, the manuscritpts should be improved.

In introduction section, the importance of the applied methods in comparision to others applied nowadays should be evidenced. Besides, english writing also should be revised.

More details on characterization of core-shell particules should be addressed.

Comparing the results obtained in this article with those obtained by other authors, what are the main gains and what is the major innovation of this work? These questions need to be answered.
After these modifications, I believe that the paper can be published in the present journal once it is robust and well structured .

Author Response

13-Oct-2021

Letter to the reviewers of Cancers

Subject: Compliance to reviewers’ comments in regards of the Manuscript # cancers-1406449.

Dear Prof. Dr. Samuel C. Mok

Editor in chief of cancers Journal

It is a great pleasure to have this chance of publishing in your respective journal “Cancers”. Herein, we are trying to respond precisely to all the valuable points suggested by reviewers and hope that it will fulfil the journal requirements and make our research deserve this opportunity to be published through this great platform.

Reviewers

I am happy to report that this version of the manuscript is significantly improved compared with the previous version I read. The relevant parts of manuscript have been updated and improved.

Finally, we would like to thank the reviewers for the constructive comments to improve our manuscript.

Thanks again for your time.

Response to Reviewer 2 Comments

Point 1: In the present paper, authores developed CD44 and folate receptor-targeting multi-func-38 tional dual drug-loaded nanoparticles (DDM). These contained a hyaluronic acid (HA) and folic 39 acid (FA) conjugated to a dichloroacetate (DCA) shell linked to a 2-deoxy glucose (2DG) and mag-40 nesium oxide (MgO) core to enhance localized chemo-radiotherapy for effective BC treatment. However well structurated and with promisor results, the manuscritpts should be improved.

Response 1: The manuscript has been improved

Point 2: In introduction section, the importance of the applied methods in comparision to others applied nowadays should be evidenced. Besides, english writing also should be revised. 

Response 2: Has been clarified in the manuscript, and the English writing has been revised

Point 3: More details on characterization of core-shell particules should be addressed.

Response 3: More details on characterization of core-shell particles had been inserted.

Point 4: Comparing the results obtained in this article with those obtained by other authors, what are the main gains and what is the major innovation of this work? These questions need to be answered.

After these modifications, I believe that the paper can be published in the present journal once it is robust and well structured. 

Response 4: The major innovation of this work in comparison with the previous study, several treatments are also still limited due to rapid relapse issues, systemic toxicity, and non-specific drug delivery, leading to low therapeutic outcomes. Also, poor survival rates in patients with BC may be attributable to radio-resistance stemming from increased cell DNA repair capabilities after RT. Our design innovation was underpinned by a novel biodegradable multi-target nanocarrier that targeted overexpressed CD44 and FR-α receptors in BC cells. This was followed by the release of three therapeutic agents for the effective chemo-radiotherapy of BC.

The main gains of this work, multi-functionalized DDM depletion-mediated metabolic reprogramming via multiple signal pathways in BC cells is a promising targeted metabolic therapy, and has been clarified in the manuscript, also these promising resultsof release kinetics of DDM under acidic conditions,indicated that DDM design can be used as oral treatment distinguished by this from drugs that injection by intravenous only. Most importantly, the combination of DDM components in a novel nano-core-shell formulation offered a better outcome as a promising antitumor drug since it allowed a superior targeting and release according to tumor associated conditions (acidic pH)

Additionally, comparing our findings with those that were performed on separate- component manner, the innovative DDM core-shell exhibited several characteristics. Firstly, it consisted of an MgO core for enhancing of a potent radio-sensitizer–2DG and DCA with a pH-sensitive degradable polymer shell for controlled release of a targeted-drug-DDM in response to acidic tumor conditions. Recently, MgO-NPs, as potential candidates for drug delivery, anticancer, magnetic resonance imaging, and hyperthermia systems have been studied. Secondly, the possible incorporation of contrast agents such as MgO into nanoparticles could help track drug uptake by TEM imaging. Thirdly, it allows for the simultaneous delivery of chemotherapeutic reagents (2DG and DA) and radio-sensitizers (e.g., MgO). Finally, HA and FA molecules on the DDM surface help target overexpressed CD44 and FR-α receptors in BC cells thereby facilitating site-specific targeted therapy.

Reviewer 3 Report

In this manuscript, Askar et al synthesized a dual targeting pH-sensitive nanoparticles for breast cancer therapy.  

I recommend Major revision before it was considered for publication:

1. It is very confusion for the term of DDM in simple summary and abstract:

In line 18, DDM stand for “ …..we developed CD44 and folate receptor-targeting multi-functional dual drug-loaded nanoparticles (DDM)”

However, it looks like DDM stand for different term : “Nanoparticles exhibited sustained DDM release properties…..” “ DDM-loaded and HA-FA-functionalized nanoparticles exhibited increased selectivity and uptake by BC cells.” 

2. In Introduction part, the authors did not discuss:

a) Why MgO nano core-shell structure is studied.

b) Why pH sensitive component is needed.

c) Components in “Release of three therapeutic agents.” However, the introduction part only discussed targeting.

3. There is no scale bar in Fig1e (SEM)

4. It is not clear what the Fig 1d and Fig.1e are representing: one nanoparticle or the aggregation of many nanoparticles?

5. The DLS in Fig.1h clearly illustrated the hydrodynamic size of 10-100nm. They are much lower than sizes from TEM and SEM. The author should clarify the results.

6. Figure 1 is too big. It is two-page long figure.

7. How are the glucose and DCA conjugated into MgO? What kind of chemistry is involved?

8. To further characterize the nanoparticles, the authors are suggested to conduct a in vivo study to demonstrate the therapeutic effect of the nanoparticles in proper cancer model.

9. Page 10, line 347: The pH-responsive release of this nanoparticles was studied based on the pH value between 3, 7 and 9, which are not appropriate for in vivo study. The pH of tumor environment is between 6.5 and 7.2.

10. Page 1, line 38: Author mentioned “pharmacotherapy”. It is not consisted with the term of chemotherapy in the late context.

11. Page 19: The FT-IR of bare MgO should be studied as control in Figure 1g. Then the absorption in 1776, 1272, 1089, 930, and 584 will be labeled and discussed.

12. The manuscript contains many typos. The author should correct the typos carefully before possible resubmission. Here are some examples:

a) Page 7, line 308: “(Figure 1g) shows the FT-IR spectra of the synthesized DDM samples.”

b) Page 10, Line 372-383: the whole paragraph is bold. It should not.

c) Page 18, line 531: “But not have been sufficient to discontinued cancer progression.” Check the sentence.

d) Page 20, line 633: “it is equally important to reduce IR mediated toxic effects in healthy tissues surrounding tumors.” It should be RT.

Author Response

13-Oct-2021

Letter to the reviewers of Cancers

Subject: Compliance to reviewers’ comments in regards of the Manuscript # cancers-1406449.

Dear Prof. Dr. Samuel C. Mok

Editor in chief of cancers Journal

It is a great pleasure to have this chance of publishing in your respective journal “Cancers”. Herein, we are trying to respond precisely to all the valuable points suggested by reviewers and hope that it will fulfil the journal requirements and make our research deserve this opportunity to be published through this great platform.

Reviewers

I am happy to report that this version of the manuscript is significantly improved compared with the previous version I read. The relevant parts of manuscript have been updated and improved.

Finally, we would like to thank the reviewers for the constructive comments to improve our manuscript.

Thanks again for your time.

Response to Reviewer 3 Comments

Point 1: It is very confusion for the term of DDM in simple summary and abstract:

Response 1: Confusion for the term of DDM has been corrected in the manuscript.

  • DDM stand for hyaluronic acid (HA) and folic acid (FA) conjugated to a dichloroacetate (DCA) shell linked to a 2-deoxy glucose (2DG) and magnesium oxide (MgO) core, and has been added in the manuscript.
  • DDM is the term for the first letters from each of dichloroacetate (DCA), 2-deoxy glucose (2DG), and magnesium oxide (MgO) (DCA@2DG@MgO), it is an abbreviated for hyaluronic acid (HA) and folic acid (FA) conjugated to a dichloroacetate (DCA) shell linked to a 2-deoxy glucose (2DG) and magnesium oxide (MgO) core, and has been added in the manuscript.

Point 2: In Introduction part, the authors did not discuss:

  • Why MgOnano core-shell structure is studied.
  • Why pH sensitive component is needed.
  • Components in “Release of three therapeutic agents.” However, the introduction part only discussed targeting.

Response 2: We discussed the loss of parts in the introduction according to reviewer recommendations.

  • Because, MgO-NPs have multi-potential activity as drug delivery, anticancer, radio-sensitization, magnetic resonance imaging and hyperthermia systems, and the incorporation of MgO-NPs in nano materials could help track drug uptake by TEM imaging and IR quantitative measurements. Also using of MgO in nano materials core can formed regularly crystal in fact and produced the face-centered cubic (fcc) crystalline configuration in nanoparticles fabricationand has been added in the manuscript.
  • Since tumor tissues constitute an acidic environment and the intracellular compartments of cancer cells (endosomes and lysosomes) provide an even lower pH than the extracellular environment (around 5.0–5.5)(1), the pH-responsive release behaviour of nanoparticles is potentially important with regard to the anticancer efficacy, and has been added in the manuscript.
  • AthinaAngelopoulou, et al. Folic Acid-Functionalized, Condensed Magnetic Nanoparticles for Targeted Delivery of Doxorubicin to Tumor Cancer Cells Overexpressing the Folate Receptor. ASC omega J 2019, 4, 22214-22227. DOI.org/10.1021/acsomega.9b03594.
  • DCA, 2DG, and MgO have been discussed in the introduction (a related part to the three components in the introduction referring to their role in the current formulation, and has been tracked in the manuscript)

Point 3: There is no scale bar in Fig1e (SEM)

Response 3: Scale bar has been added on SEM figure in the manuscript according to your recommendation.

Point 4: It is not clear what the Fig 1d and Fig.1e are representing: one nanoparticle or the aggregation of many nanoparticles?

Response 4: For more explanation, it must be noted that we must prepare solid powder in sample preparation (in both SEM and SEM/EDX mapping techniques). In this case, the aggregation may be performed, so the particles appear in figures 1d and 1e as aggregated particles with a bright core and condensed organic shell.

Point 5: The DLS in Fig.1h clearly illustrated the hydrodynamic size of 10-100nm. They are much lower than sizes from TEM and SEM. The author should clarify the results.

Response 5: For more explanation, it must be noted that we must prepare solid powder in sample preparation (in both SEM and TEM techniques). In this case, the aggregation may be performed, so the particles appear in SEM and TEM figures as aggregated particles with a bright core and condensed organic shell (SEM), and vice versa in the case of TEM images, so the average particle size may be higher than DLS due to aggregation. In contrast, the particles appear separated in DLS measurements because the sample must be ultrasonicated in solution, and the particle size distribution may be lower than the particle size. In the typical cases, the particle size distribution determination in DLS must be higher than the average particle size determination by TEM due to the hydrodynamic radius formed between the solution and the formed nanocomposites.

Point 6: Figure 1 is too big. It is two-page long figure.

Response 6: Figure 1 has been divided into two figures in the manuscript according to your recommendation.

Point 7: How are the glucose and DCA conjugated into MgO? What kind of chemistry is involved?

Response 7: The preparation method is impregnation method depending on the weak van der waal forces for the easy dissociation of each layer of the synthesized nanocomposite and for better performance against cancer cells.

Point 8: To further characterize the nanoparticles, the authors are suggested to conduct a in vivo study to demonstrate the therapeutic effect of the nanoparticles in proper cancer model.

Response 8: This study was designed to promptly address the major cytotoxic profile of the novel fabricated nano-core-shell platform for investigating the efficacy of innovative targeted formulation based on 2DG@DCA@MgO against certain types of breast cancer cell lines; MCF-7 & MDA-MB-231, it is therefore, the findings obtained in the current study will be utilized in expanding our aim to include in vivo study in a future perspective. To the best of our knowledge, in vitro studies are used as a verified preliminary data guiding to study the drug in vivo. Considering that DDM is applied for the first time, we believe that in vitro study is the most proper mean of evaluation before the transfer to in vivo studies. This study covered the major points of the DDM anticancer effects in vitro as compared with normal and untreated cells, indicating that DDM enhanced selectivity towards the breast cancer cells devoid of apparent cytotoxic effects against normal cells (MCF-10A) or vehicle- treated cells. Various previous studies have reported the newly formulated nanoparticles as a promising anticancer drugs through studies based on in vitro data(1- 3). Hopefully, the current findings will be used in a further study conducting in vivo model as per reviewer recommendation.

  • Chan L, et al. Cancer-Targeted Selenium Nanoparticles Sensitize Cancer Cells to Continuous γ Radiation to Achieve Synergetic Chemo-Radiotherapy. Chem Asian J. 2017 Dec 5;12(23):3053-3060. doi: 10.1002/asia.201701227.
  • AthinaAngelopoulou, et al. Folic Acid-Functionalized, Condensed Magnetic Nanoparticles for Targeted Delivery of Doxorubicin to Tumor Cancer Cells Overexpressing the Folate Receptor. ASC omega J 2019, 4, 22214-22227. DOI.org/10.1021/acsomega.9b03594.
  • Markowski A, et al. Evaluation of the In Vitro Cytotoxic Activity of Ursolic Acid PLGA Nanoparticles against Pancreatic Ductal Adenocarcinoma Cell Lines. Materials (Basel). 2021 Aug 29;14(17):4917. doi: 10.3390/ma14174917.

Point 9: Page 10, line 347: The pH-responsive release of this nanoparticles was studied based on the pH value between 3, 7 and 9, which are not appropriate for in vivo study. The pH of tumor environment is between 6.5 and 7.2.

Response 9: Generally, The pH of tumor environment is between 5-6.9.(1)  Since the tumor tissues constitute an acidic environment and the intracellular compartments of cancer cells (endosomes and lysosomes) provide an even lower pH than the extracellular environment (around 5.0–5.5).(2) It is worth mentioning that the encapsulation system of the nanoparticles maintain their structural integrity at 1.5 pH, and the contents can only be released slowly at pH > 5, which suggests the stability of the encapsulation system under the conditions of simulated stomach, and release the contents slowly under the conditions of intestinal tract.(1) These promising results indicated that DDM design can be used  via different routes of administration with a pronounced stability under a wide range of pH (3- 9 pH) as revealed in our data. Accordingly, the novel nano-core-shell targeted DDM formulation possesses anticancer characteristic with enhanced targeting towards breast cancer cells and apparent stability with maintained patient compliance later on. Worthwhile, the simultaneous release of the active components (2DG, DCA, and MgO) takes place within the tumor microenvironment (at pH 3 and upwards to approximately pH 5) after the active targeting via the outer shell annex (HA and FA).   The requested clarification has been included in the manuscript according to your recommendation.

  • Zhuo S, et al. pH-Sensitive Biomaterials for Drug Delivery. Molecules2020, 25, 5649. DOI:10.3390/molecules25235649
  • AthinaAngelopoulou, et al. Folic Acid-Functionalized, Condensed Magnetic Nanoparticles for Targeted Delivery of Doxorubicin to Tumor Cancer Cells Overexpressing the Folate Receptor. ASC omega J2019, 4, 22214-22227.org/10.1021/acsomega.9b03594

Point 10: Page 1, line 38: Author mentioned “pharmacotherapy”. It is not consisted with the term of chemotherapy in the late context.

Response 10: Has been corrected in the manuscript.

Point 11: Page 19: The FT-IR of bare MgO should be studied as control in Figure 1g. Then the absorption in 1776, 1272, 1089, 930, and 584 will be labeled and discussed.

Response 11: Done, a new FTIR analysis regarding bare MgO NPs had been performed, compared with the synthesized nanocomposite, and discussed in details as the following:

After conducting a comparative FTIR analysis of bare MgO NPs, a peak located at 3230 cm-1 was corresponded to the -OH stretching region, also another peak located at 617 cm-1 was assigned to the stretching mode of the Mg-O core, which slightly shifted as compared with Mg-O (680 cm-1) in the synthesized nanocomposite due to the absence of organic shells.

Point 12: The manuscript contains many typos. The author should correct the typos carefully before possible resubmission. Here are some examples:

  • a) Page 7, line 308: “(Figure 1g) shows the FT-IR spectra of the synthesized DDM samples.”
  • Page 10, Line 372-383: the whole paragraph is bold. It should not.
  • Page 18, line 531: “But not have been sufficient to discontinued cancer progression.” Check the sentence.
  • Page 20, line 633: “it is equally important to reduce IR mediated toxic effects in healthy tissues surrounding tumors.” It should be RT.

Response 12: The typos in the manuscript have been corrected according to your recommendation.

  • Has been corrected in the manuscript.
  • Has been corrected in the manuscript.
  • Has been clarified in the manuscript.
  • Has been corrected in the manuscript.

Round 2

Reviewer 1 Report

  1. Inappropriate data presentation in Figures 3f and 4d. Take Figure 3f as an example, you can not combine different cell lines as a group. You should present the data from each cell line with different treatments separately.  
  2. About the in vivo study: An experiment is worth a thousand words.

Author Response

21-Oct-2021

Letter to the reviewers of Cancers

Subject: Compliance to reviewers’ comments in regards of the Manuscript # cancers-1406449.

Dear Prof. Dr. Samuel C. Mok

Editor in chief of cancers Journal

It is a great pleasure to have this chance of publishing in your respective journal “Cancers”. Herein, we are trying to respond precisely to all the valuable points suggested by reviewers and hope that it will fulfil the journal requirements and make our research deserve this opportunity to be published through this great platform.

Reviewers

I am happy to report that this version of the manuscript is significantly improved compared with the previous version I read. The relevant parts of manuscript have been updated and improved.

Finally, we would like to thank the reviewers for the constructive comments to improve our manuscript.

Thanks again for your time.

Response to Reviewer 1 Comments

Point 1: Inappropriate data presentation in Figures 3f and 4d. Take Figure 3f as an example, you can not combine different cell lines as a group. You should present the data from each cell line with different treatments separately.

Response 1: Has been corrected in the manuscript as per reviewer recommendation.

Point 2: About the in vivo study: An experiment is worth a thousand words.

Response 2: The authors would like to acknowledge the reviewer’s valuable opinion and suggestion. The authors entirely agree with the reviewer that in vivo studies strongly support the in vitro data. Hopefully, the current findings will be utilized in expanding our aim to include in vivo study in a future perspective as per reviewer recommendation.

Reviewer 2 Report

Dear all,

After reading the revised version of this manuscript, I could observe that the corrections and suggestions were taken into account, and together with the observations made by the other referees, the final file presented great improvement and made the final paper more attractive and interesting.

In my opinion, this manuscript is now ready for acceptance into the present journal. 

Author Response

21-Oct-2021

Letter to the reviewers of Cancers

Subject: Compliance to reviewers’ comments in regards of the Manuscript # cancers-1406449.

Dear Prof. Dr. Samuel C. Mok

Editor in chief of cancers Journal

It is a great pleasure to have this chance of publishing in your respective journal “Cancers”. Herein, we are trying to respond precisely to all the valuable points suggested by reviewers and hope that it will fulfil the journal requirements and make our research deserve this opportunity to be published through this great platform.

Reviewers

We would like to thank the reviewer for the constructive comments to improve our manuscript and his opinion for acceptance of this manuscript in the cancers journal.

Thanks again for your time.

Response to Reviewer 2 Comments

We would like to thank the reviewer for the constructive comments to improve our manuscript and his opinion for acceptance of this manuscript in the cancers journal.

Reviewer 3 Report

  1. There is big concern for synthesis of DDM.

The DDM is synthesized through several steps. The general procedure is mixing and centrifuge. The authors believe the synthesis is based on weak van der waal forces. It is not reasonable that van der waal can incorporate DCA, 2DG, HA and FA layer by layer. Usually, covalent bond is needed to conjugate the targeting component. In addition, element map is not enough to track the incorporation of DCA, 2DG, HA, FA

  1. There are some design issues:

The author explained that “encapsulation system of the nanoparticles maintain their structural integrity at 1.5 pH, and the contents can only be released slowly at pH > 5, which suggests the stability of the encapsulation system under the conditions of simulated stomach”

If the DDM is designed for oral administration, the targeting purpose of HA and FA will not be realized since the DDM will be destroyed by strong acid in stomach.   

If the DDM will be administered via an IV administration, it looks like the DCA and 2DG are released anywhere because pH in normal tissue is even higher than that of tumor.   

Author Response

21-Oct-2021

Letter to the reviewers of Cancers

Subject: Compliance to reviewers’ comments in regards of the Manuscript # cancers-1406449.

Dear Prof. Dr. Samuel C. Mok

Editor in chief of cancers Journal

It is a great pleasure to have this chance of publishing in your respective journal “Cancers”. Herein, we are trying to respond precisely to all the valuable points suggested by reviewers and hope that it will fulfil the journal requirements and make our research deserve this opportunity to be published through this great platform.

Reviewers

I am happy to report that this version of the manuscript is significantly improved compared with the previous version I read. The relevant parts of manuscript have been updated and improved.

Finally, we would like to thank the reviewers for the constructive comments to improve our manuscript.

Thanks again for your time.

Response to Reviewer 3 Comments

Point 1: There is big concern for synthesis of DDM.

The DDM is synthesized through several steps. The general procedure is mixing and centrifuge. The authors believe the synthesis is based on weak van der waal forces. It is not reasonable that van der waal can incorporate DCA, 2DG, HA and FA layer by layer. Usually, covalent bond is needed to conjugate the targeting component. In addition, element map is not enough to track the incorporation of DCA, 2DG, HA, FA

Response 1: It Done, thank you for the reviewer opinion and recommendation regarding the types of chemical bonds between core-shell structures. We apologies for the only physical al bond illustration in the last revision. After the literature comparison achieved between the FTIR data of bare FA, HA, DCA, 2DG, and bare MgO-NPs (data presented in Fig. 2C), it must be noted that the connection type between the outer organic shells was by covalent bond due to the presence of new peaks as describes before which not present in bare FA, HA, DCA, 2DG, and bare MgO NPs which indicated the conjugation behaviour between layers and indicated by a covalent bond as described in the recent publications. Our FTIR results detected the conjugation behaviour as new peaks formed (strong covalent bond) or a minor shifting in the bare peaks (weak physical bond). Our responses had been supported with recent references and red highlighted in the revised manuscript.

Point 2: The author explained that “encapsulation system of the nanoparticles maintain their structural integrity at 1.5 pH, and the contents can only be released slowly at pH > 5, which suggests the stability of the encapsulation system under the conditions of simulated stomach”

If the DDM is designed for oral administration, the targeting purpose of HA and FA will not be realized since the DDM will be destroyed by strong acid in stomach.  

If the DDM will be administered via an IV administration, it looks like the DCA and 2DG are released anywhere because pH in normal tissue is even higher than that of tumor.

Response 2: It is widely accepted that the pH of cancer cells is much more acidic than normal cells. Generally, pH values of the normal tissues (brain tissues, subcutaneous tissues, etc.) are in the range of 7.2–7.5. However, the pH of tumor cells is mildly acidic in the range of 5–6.9, and the intracellular compartments of cancer cells (endosomes and lysosomes) provides an even lower pH than the extracellular environment (around 5.0–5.5).(1,2,3)

It is worth mentioning that the accepted encapsulation system of the nanoparticles maintains their structural integrity at 1.5 pH (stomach pH),(1) and the contents can only be released slowly at pH > 3, which suggests the stability of the encapsulation system under the conditions of the simulated stomach conditions and release the nanocomposite contents slowly under the pH related conditions and environment of the intestinal tract, which is less acidic than the stomach condition and tend to be weak acidic to slightly alkaline (pH 6- 7.4). Worthwhile, our results are compatible with the accepted nanoparticles formulations in the previous studies.(1,2,3) Our results showed that the simultaneous release of the active components (2DG, DCA, and MgO) takes place within conditions similar to the tumor microenvironment (at pH 3 and upwards to approximately pH 5) after the active targeting via the outer shell annex (HA and FA). With respect to the concerns about the formulation destruction after oral administration, the authors would like to reassure the reviewer apprehension toward this point that can be easily overcome in the future through providing an outer enteric- coating like those used to protect acid- labile drugs that permits the dissolution of these coating only under the higher pH in the intestinal tracts (for example, the enteric coating with hydroxypropyl methylcellulose phthalate (HP55), which could withstands acidic pH in the stomach and later selectively dissolves in the pH of the intestine protecting our formulation from spoiling and unintended release. The requested clarification has been included in the manuscript according to your recommendation.

  • Zhuo S, et al. pH-Sensitive Biomaterials for Drug Delivery. Molecules2020, 25, 5649. DOI:10.3390/molecules25235649
  • Guanyu Hao, et al. Manipulating extracellular tumour pH: An effective target for cancer therapy. RSC Advances 2018, 8(39):22182-22192. DOI:1039/C8RA02095G
  • AthinaAngelopoulou, et al. Folic Acid-Functionalized, Condensed Magnetic Nanoparticles for Targeted Delivery of Doxorubicin to Tumor Cancer Cells Overexpressing the Folate Receptor. ASC omega J2019, 4, 22214-22227.org/10.1021/acsomega.9b03594

Round 3

Reviewer 1 Report

No other comment. Good luck with your animal experiments.

Author Response

25-Oct-2021

Letter to the reviewers of Cancers

Subject: Compliance to reviewers’ comments in regards of the Manuscript # cancers-1406449.

Dear Prof. Dr. Samuel C. Mok

Editor in chief of cancers Journal

It is a great pleasure to have this chance of publishing in your respective journal “Cancers”. Herein, we are trying to respond precisely to all the valuable points suggested by reviewers and hope that it will fulfil the journal requirements and make our research deserve this opportunity to be published through this great platform.

Reviewers

We would like to thank the reviewer for the constructive comments to improve our manuscript and his opinion for acceptance of this manuscript in the cancers journal.

Thanks again for your time.

Response to Reviewer 1 Comments

We would like to thank the reviewer for the constructive comments to improve our manuscript and his opinion for acceptance of this manuscript in the cancers journal.

Reviewer 3 Report

publish after corrections to minor methodological errors and text editing

Author Response

25-Oct-2021

Letter to the reviewers of Cancers

Subject: Compliance to reviewers’ comments in regards of the Manuscript # cancers-1406449.

Dear Prof. Dr. Samuel C. Mok

Editor in chief of cancers Journal

It is a great pleasure to have this chance of publishing in your respective journal “Cancers”. Herein, we are trying to respond precisely to all the valuable points suggested by reviewers and hope that it will fulfil the journal requirements and make our research deserve this opportunity to be published through this great platform.

Reviewers

I am happy to report that this version of the manuscript is significantly improved compared with the previous version I read. The relevant parts of manuscript have been updated and improved.

Finally, we would like to thank the reviewers for the constructive comments to improve our manuscript.

Thanks again for your time.

Response to Reviewer 3 Comments

Point 1: publish after corrections to minor methodological errors and text editing

Response 1: Editing language and the errors in the manuscript have been done as peer reviewer recommendation.